# Suppression of ERK signalling promotes pluripotent epiblast in the human blastocyst

Claire S. Simon [1,2], Afshan McCarthy[2], Laura Woods[1], Desislava Staneva [1], Martin Proks[3], Nazmus Salehin [3], Georgia Lea [1], Qiulin Huang[1,2], Madeleine Linneberg-Agerholm[3], Alex Faulkner[4], Athanasios Papathanasiou [5], Kay Elder [5], Phil Snell[5], Leila Christie[5], Patricia Garcia[6], Valerie Shaikly[6], Mohamed Taranissi[6], Meenakshi Choudhary[4], Mary Herbert[4,7], Courtney W. Hanna [1], Joshua M. Brickman [3] & Kathy K. Niakan [1,2,8,9] ✉

Studies in the mouse demonstrate the importance of fibroblast growth factor (FGF) and extra-cellular receptor tyrosine kinase (ERK) in specification of embryo-fated epiblast and yolk-sac-fated hypoblast cells from uncommitted inner cell mass (ICM) cells prior to implantation. Molecular mechanisms regulating specification of early lineages in human development are comparatively unclear. Here we show that exogenous FGF stimulation leads to expanded hypoblast molecular marker expression, at the expense of the epiblast. Conversely, we show that specifically inhibiting ERK activity leads to expansion of epiblast cells functionally capable of giving rise to naïve human pluripotent stem cells. Single-cell transcriptomic analysis indicates that these epiblast cells downregulate FGF signalling and maintain molecular markers of the epiblast. Our functional study demonstrates the molecular mechanisms governing ICM specification in human development, whereby segregation of the epiblast and hypoblast lineages occurs during maturation of the mammalian embryo in an ERK signal-dependent manner.

Formation of the blastocyst is a critical event in embryogenesis occurring in the first week of human development. Two sequential cell fate decisions segregate the embryonic pluripotent epiblast from the extra-embryonic tissues, trophectoderm and hypoblast. Errors in cell differentiation can cause embryo arrest, leading to miscarriage. More than half of all natural conceptions are estimated to end in very early (<5 week) pregnancy loss[1]. Despite the significance for human health and stem cell biology, we do not understand the mechanisms which direct cell differentiation in the early human embryo. To improve our understanding of the mechanisms of cell fate specification in the early embryo, we studied the effects of cell-cell communication during this critical window of development.

The ability of FGFs to bias ICM cells towards hypoblast is conserved amongst mammals studied to date, including in rodents, rabbits, bovine and pig[2–5]. The molecular pathway governing ICM cell fate segregation has been elucidated genetically in the mouse. Within mouse ICM cells, and later in epiblast cells, the pluripotency marker NANOG induces *Fgf4* expression[6]. FGF4 ligand initially signals in an

[1]Loke Centre for Trophoblast Research, Department of Physiology, Development and Neuroscience, University of Cambridge, Cambridge CB2 3EG, UK. [2]Human Embryo and Stem Cell Laboratory, The Francis Crick Institute, 1 Midland Road, London NW1 1AT, UK. [3]Novo Nordisk Foundation Center for Stem Cell Medicine (reNEW), Department of Biomedical Science, University of Copenhagen, Copenhagen, Denmark. [4]Newcastle Fertility Centre, Newcastle upon Tyne Hospitals NHS Trust, Biosciences Institute, Newcastle University Centre for Life, Newcastle upon Tyne, UK. [5]Bourn Hall Clinic, Bourn, Cambridge CB23 2TN, UK. [6]Assisted Reproduction and Gynaecology Centre, London W1G 6LP, UK. [7]Department of Anatomy & Developmental Biology, Biomedicine Discovery Institute, Monash University, Melbourne, Australia. [8]Wellcome Trust – Medical Research Council Stem Cell Institute, University of Cambridge, Jeffrey Cheah Biomedical Centre, Puddicombe Way, Cambridge CB2 0AW, UK. [9]Epigenetics Programme, Babraham Institute, Cambridge CB22 3AT, UK. ✉e-mail: kkn21@cam.ac.uk

autocrine manner to uncommitted ICM progenitors predominantly via FGFR1, and subsequently by FGFR2 in hypoblast precursors[7,8]. Signalling activation specifies hypoblast via a GRB2/MEK/ERK cascade leading to an upregulation of hypoblast specific markers such as *Gata6*, and a downregulation of epiblast markers like *Nanog*[9–11]. These molecular insights have also informed strategies to establish naïve mouse pluripotent stem cells in vitro that more closely resemble the blastocysts stage epiblast in vivo from which they were derived[12,13]. Paradoxically, in human embryos, previous studies suggested that upstream FGF receptor (FGFR) or mitogen-activated protein kinase (MEK) inhibition did not affect hypoblast formation[3,14].

We recently demonstrated crosstalk of FGF-driven mitogen-activated protein kinase (MEK) and insulin growth factor-driven phosphoinositol-3 kinase (PI3K) activity upstream of ERK signalling in human pluripotent stem cells (hPSCs) derived from the epiblast[15]. In addition, we showed that hypoblast differentiation in vitro from naïve hPSCs depends on FGF signalling and that naïve hPSCs can be maintained by simultaneously blocking the FGF receptor and its downstream kinase MEK[16]. We hypothesized that ERK may have a conserved role in human embryo hypoblast versus epiblast specification.

Here, we determined that exogenous FGF signalling activity is sufficient to specify the hypoblast in human blastocysts. We describe ERK signalling during the blastocyst stage and demonstrate that blocking ERK activity leads to expansion of epiblast cells functionally capable of giving rise to naïve hPSCs. Transcriptomic analysis further reveals that these epiblast cells downregulate FGF signalling, while maintaining molecular markers of the naïve epiblast. Our study provides mechanistic insights into how ERK inhibition affects lineage progression across different cell types in the early human embryo, including the ectopic expression of pluripotency markers in non-epiblast lineages. We reveal species-specific differences in ERK function, with human ERK-inhibited epiblast retaining naïve pluripotency while mouse ERK-inhibited epiblast exhibits a dormant pluripotent state. Our functional studies provide mechanistic insight into human blastocyst formation and reveal the molecular mechanisms that regulate ICM specification in humans. We propose a unified model in which segregation of the epiblast and hypoblast lineages occurs during maturation of the mammalian blastocyst in an ERK signal-dependent manner.

## Results

### Exogenous FGF is sufficient to drive human hypoblast specification

We hypothesized that FGF4 may play a conserved role in human preimplantation development. Transcriptional analysis of human preimplantation embryos[17–19] indicates that there is pan-ICM expression of the FGFR1 receptor, while FGF4 is expressed specifically in epiblast cells (Supplementary Data Fig. 1a), consistent with current mouse models of ICM segregation where FGF4 acts predominantly via FGFR1 upstream of pERK in uncommitted ICM cells to specify hypoblast[7,8].

To test the effect of exogenous FGF signalling on human embryos we carried out a dose-response experiment. We treated Day 5 human embryos and cultured them for 36 h in medium containing 0 ng/ml, 250 ng/ml, 500 ng/ml, and 750 ng/ml FGF4 plus Heparin (which stabilizes FGF interactions) (Fig. 1a). Immunofluorescence analysis of NANOG (epiblast), GATA4 (hypoblast) and GATA3 (trophectoderm, placenta progenitor cells) showed that increasing concentrations of FGF4 altered cell fate specification in the ICM (Fig. 1a–c and Supplementary Data Fig. 1b–f). In comparison to controls, the 750 ng/ml FGF4 treated embryos had a 1.5-fold increase in hypoblast cells (Fig. 1b, mean 12 vs 8 hypoblast cells per embryo, $p = 0.21$) (Fig. 1b), while showing a significant 2-fold reduction in epiblast cells (Fig. 1c, mean 3 vs 8 epiblast cells per embryo $p = 0.015$). Increasing concentrations of FGF4 had no significant effect on ICM, trophectoderm or total cell number (Supplementary Data Fig. 1c–e). Consistent with changes in both epiblast and hypoblast

numbers, we found that increasing concentration of FGF4 increased the ratio of hypoblast: epiblast cells in the ICM (Fig. 1d, Control 39% hypoblast, 250 ng/ml 60%, $p = 0.030$, 500 ng/ml 73%, $p = 0.0016$, 750 ng/ml 75%, $p = 0.0014$), and embryos exhibited an all-hypoblast ICM in a dose-dependent manner (Supplementary Data Fig. 1f).

This is consistent with the response in mouse and cow embryos to FGF4, where NANOG or SOX2 (epiblast) expression is downregulated and the ICM is predominantly comprised of SOX17- or GATA6-expressing hypoblast cells (Supplementary Data Fig. 1g, f), similar to previous studies[2,3]. Moreover, we determined that rat preimplantation embryos also responded to FGF4, downregulating NANOG and upregulating GATA6 expression throughout the ICM (Supplementary Data Fig. 1h). Overall, our findings demonstrate that FGF4 is sufficient to drive human hypoblast specification, a mechanism conserved across species.

### Suppression of ERK signalling blocks hypoblast formation in the human blastocyst

FGFs can activate multiple downstream pathways, including ERK, PKC and PI3K[20]. Our previous work identified active phosphorylated ERK (pERK) in human whole blastocyst protein lysates[15]. However, we had not defined the cell type and stage that contains the highest signalling activity. To address this question, we analyzed pERK by immunofluorescence staining[21], beginning with mouse embryos to develop methodology to preserve phosphorylation of ERK protein during fixation. Consistent with previous studies[21,22], pERK is detectable in the ICM (Supplementary Data Fig. 2a). We next characterized pERK by immunofluorescence analysis in primed human embryonic stem cells (hESC) and detected pERK cytoplasmic localization. As expected, inhibition of the upstream kinase, MEK, using PD0325901 led to downregulation of pERK in hESC, indicating specificity of the immunofluorescence analysis (Supplementary Data Fig. 2b). Next, we stained for pERK in human blastocysts from Days 5 to 6.5, using SOX2 and OTX2 of the epiblast and hypoblast, respectively, in these earlier staged embryos (Fig. 2a). Active pERK is evident in the cytoplasm of epiblast, hypoblast and trophectoderm cell types, with a high proportion of hypoblast having pERK[high] cells (Fig. 2b). The level of pERK in embryos increases over time (Fig. 2c), altogether suggesting a role for this signalling pathway in early human development.

To determine if ERK is the effector of ICM specification, we used Ulixertinib[23], a selective ATP-competitive inhibitor of ERK1/2 (referred to as ERKi). As expected, ERKi led to downregulation of GATA4 hypoblast in the mouse and an ICM exclusively comprised of SOX2 expressing epiblast cells (Supplementary Data Fig. 2c), consistent with previous studies[24]. In addition, the ICM of cow embryos following ERKi lost GATA6+ hypoblast cells (Supplementary Data Fig. 2d).

We then cultured Day 5 human embryos for 36 h in medium containing volume matched DMSO (Control) or 5 μM Ulixertinib (ERKi) (Fig. 2d) followed by quantitative immunofluorescence staining (Supplementary Data Fig. 2e). Blocking ERK signalling resulted in embryos with no GATA4 (hypoblast) expression, and an ICM comprising of predominantly NANOG positive (epiblast) cells (Fig. 2d–f). ERKi treatment thus led to a loss of hypoblast in most treated embryos (Fig. 2e, $p = 0.003$, 2 vs 13 hypoblast cells per embryo). Additionally, there was a modest increase in the number of epiblast cells in ERKi embryos compared with controls (Fig. 2f, $p = 0.4$, 15 vs 12 epiblast cells per embryo). The proportion of hypoblast: epiblast cells was dramatically distorted in ERKi treated embryos as there were no significant changes in ICM cell numbers (Fig. 2g, Supplementary Data Fig. 2f, g, control 52% hypoblast, and ERKi 9% hypoblast, $p = 0.0002$). The majority of ERKi embryos demonstrated an all-epiblast ICM phenotype (Supplementary Data Fig. 2g), while those that retained some hypoblast cells also had a greater cell number indicating a more advanced developmental age, and suggesting this residual hypoblast could be a result of lineage specification that occurred prior to exposure to ERKi.

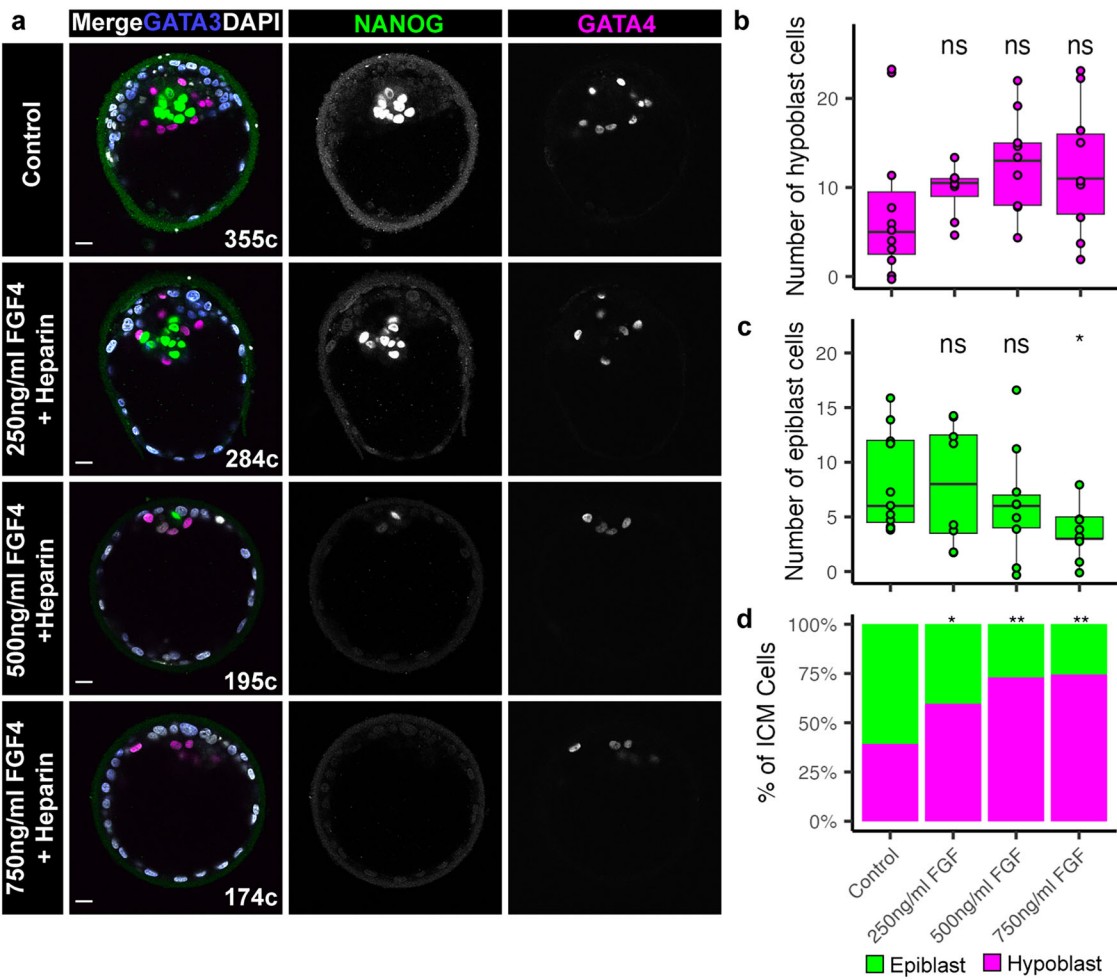

**Fig. 1 | Exogenous FGF is sufficient to drive human hypoblast specification.**
**a** Confocal images of Day 6.5 human embryos immunofluorescently labelled for lineage markers NANOG (epiblast), GATA4 (hypoblast) and GATA3 (trophectoderm), and stained for nuclear DAPI. Human embryos were cultured in Control medium, or medium supplemented with increasing concentrations of FGF and Heparin as indicated from Day 5 for 36 h. Total cell number (**c**) indicated. Scale bars 20 μm. **b–c** Boxplots showing the number of (**b**) hypoblast (GATA4 + NANOG-GATA3-) and (**c**) epiblast (NANOG + GATA4-GATA3-) cells in human embryos cultured in increasing concentrations of FGF and Heparin. Boxplots represent the interquartile (IQR range), with the median shown as a central line; whiskers extend to lowest or highest value within 1.5 * IQR; values for each embryo are shown as individual points. Control $n = 11$, 250 ng/ml $n = 8$, 500 ng/ml $n = 9$, 750 ng/ml $n = 9$. **d** Stacked bar charts showing the mean proportion of epiblast and hypoblast in the ICMs per embryo in each treatment group. Control $n = 11$, 250 ng/ml $n = 8$, 500 ng/ml $n = 9$, 750 ng/ml $n = 9$. Two-tailed $t$-test, n.s. not significant * $p < 0.05$, ** $p < 0.01$, *** $p < 0.001$. Source data are available on Github.

We confirmed this shift in ICM cell fate specification with SOX2 (epiblast) and PDGFRA (hypoblast) markers, where upon ERKi ICM cells lost PDGFRA expression (Supplementary Data Fig. 2h). We also observed a similar phenotype when culturing embryos for a shorter window of 24 h, from Day 5 in ERKi. Here, ERKi embryos down-regulated the early hypoblast marker GATA6 and retained high levels of pluripotency epiblast markers NANOG and OCT4 (Supplementary Data Fig. 2i). Thus, human ICM specification to epiblast and hypoblast likely occurs between the early- (Day 5) to mid- (Day 6) blastocyst stages. Together, these results show ERK signalling is active at the time of ICM segregation in human blastocysts, and suppression of this pathway blocks hypoblast formation.

ERKi treated embryos form hatching, expanded blastocysts by Day 6.5, with GATA3+ trophectoderm (Fig. 2d), indicating that formation of the trophectoderm, unlike the hypoblast, is not dependent on ERK signalling. However, ERKi embryos do have significantly fewer number of trophectoderm cells than controls (Supplementary Data Fig. 2j, $p = 0.04$) reducing the overall total embryo cell numbers (Supplementary Data Fig. 2k), suggesting a conserved role of ERK signalling in the proliferation of the human trophectoderm, similar to mouse[12].

## ERKi of human embryos alters lineage specification and maintains naïve pluripotency characteristics

To characterize the impact of loss of ERK, we performed single-cell RNA-seq analysis on ERKi and untreated control human embryos. After quality control filtering, we retained 53 cells from 6 ERKi treated embryos, and 36 cells from 5 control embryos. Dimensionality reduction followed by gene expression analysis, clustered the cells into three distinct populations, which were assigned cell types based on expression of lineage markers *NANOG, SOX2, KLF17, TDGF1* (epiblast), *GATA4, SOX17, PDGFRA, GATA6* (hypoblast) and *GATA2, GATA3, KRT18, TEAD3* (trophectoderm) (Fig. 3a–c, Supplementary Data Fig. 3a). The proportion of epiblast cells in the single-cell RNAseq dataset was enriched in the ERKi treatment (Supplementary Data Fig. 3b), supporting our findings that ERKi treatment leads to a switch of ICM cell fate toward the pluripotent epiblast (Fig. 2g).

Differential gene expression analysis between ERKi and control epiblast cells showed several misregulated genes (Fig. 3d). Transcripts for the pan-ICM FGF receptor (*FGFR1*), ERK-regulated ETS transcription factors (*ETV1, ETV4,* and *ETV5*) were downregulated following ERKi treatment (Fig. 3e, Supplementary Data 1). Whereas, naïve-pluripotency related genes (*KLF2, KLF5* and *DNMT3L*) were upregulated in ERKi

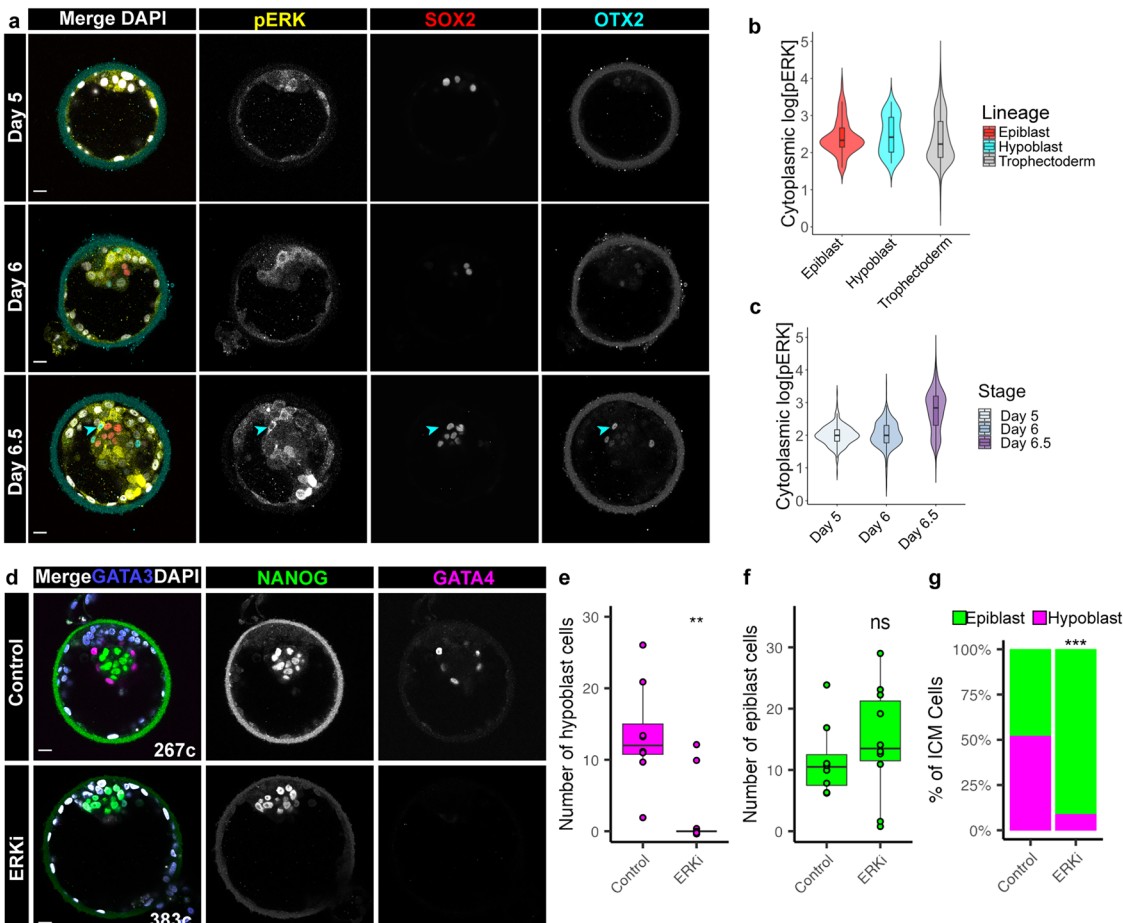

**Fig. 2 | Suppression of ERK signalling blocks hypoblast formation in the human blastocyst. a** Confocal images of Day 5 – 6.5 human embryos immuno-fluorescently labelled for phosphorylated (p)-ERK, lineage markers SOX2 (epiblast), OTX2 (hypoblast), and stained for nuclear DAPI. Cyan arrow indicates a hypoblast cell with high pERK levels. Scale bars 20 μm. **b** Violin plots showing the cytoplasmic fluorescence intensity of pERK in each lineage in embryos shown in (**a**). Boxplots represent the interquartile (IQR range), with the median shown as a central line; whiskers extend to lowest or highest value within 1.5 * IQR. Epiblast $n = 83$ cells, hypoblast $n = 41$ cells, trophectoderm $n = 1535$ cells; $n = 10$ embryos. **c** Violin plots showing the cytoplasmic fluorescence intensity of pERK over time in embryos shown in (**a**). Boxplots represent the interquartile (IQR range), with the median shown as a central line; whiskers extend to lowest or highest value within 1.5 * IQR. Day 5 $n = 313$ cells; 3 embryos, Day 6 $n = 542$ cells; 3 embryos, Day 6.5

$n = 804$ cells; 4 embryos. **d** Confocal images of Day 6.5 human embryos immu-nofluorescently labelled for lineage markers NANOG (epiblast), GATA4 (hypoblast) and GATA3 (trophectoderm) and stained for nuclear DAPI. Human embryos cultured with or without ERKi (5 μm Ulixertinib) from Day 5 for 36 h. Total cell number (**c**) indicated. Scale bars 20 μm. **e, f** Boxplots showing the number of (**e**) hypoblast (GATA4 + ) and (**f**) epiblast (NANOG + ) cells in human embryos cultured with and without ERKi. Boxplots represent the interquartile (IQR range), with the median shown as a central line; whiskers extend to lowest or highest value within 1.5 * IQR; values for each embryo are shown as individual points. Control $n = 8$, ERKi $n = 10$. **g** Stacked bar charts showing the mean proportion of epiblast and hypoblast in the ICMs per embryo in each treatment group. Control $n = 8$, ERKi $n = 10$. Two-tailed $t$-test, ns = not significant, $*p < 0.05$, $**p < 0.01$, $***p < 0.001$. Source data are available on Github.

epiblast cells. Among the differentially expressed genes, the top enriched gene sets; oestrogen response, mTOR signalling, and cholesterol and fatty acid metabolism, were all upregulated in ERKi epiblast cells (Supplementary Data Fig. 3c). These biological processes are related to lipid metabolism, a hallmark of naïve pluripotency in human ESCs and embryos[25], and suggest a shift in bioenergetic requirements of ERKi epiblast to a more naïve-like pluripotent state.

We observed a small number of hypoblast cells in ERKi treatments (Supplementary Data Fig. 3b), like our immunostaining experiments (Fig. 2g), which may have been specified prior to the start of treatment. To understand the developmental trajectory of specified hypoblast progenitors upon ERKi, we performed differential gene expression (Supplementary Data Fig. 3d). Although ERKi cells expressed some some hypoblast markers (Fig. 3c, Supplementary Data Fig. 3a), they exhibited lower levels of other hypoblast marker genes (*BMP2, OTX2* and *LGALS2*)[26] compared to control hypoblast cells within the same cluster. They also showed reduced expression of hypoblast-related FGF response genes (*DUSP4-7, SPRY4,* and *FGFR2*)[21], while retaining

*NANOG* transcription, indicating defective hypoblast cell fate commitment (Supplementary Data Fig. 3e). In addition, there were mis-regulated genes in common with epiblast cells, including *ETV5* and *FGFR1* (downregulated), and *DNMT3L* (upregulated) (Fig. 3e, and Supplementary Data Fig. 3e) indicating a shared ERK-dependent ICM-gene regulatory network.

By contrast, few genes were differentially expressed between treated and untreated trophectoderm (Supplementary Data Fig. 3f), indicating that although ERK regulates trophectoderm proliferation, it does not have a large effect on the transcription of gene-regulatory networks in this lineage (Supplemental Data Fig. 3d). *NANOG* is also ectopically expressed in the ERKi trophectoderm (Supplementary Data 1), similar to ERKi hypoblast, suggesting that ERK activity in extra-embryonic lineages may be required for the repression of naïve pluripotency, thereby helping to re-enforce an embryonic vs extra-embryonic identity.

Next, we integrated our dataset to a compiled reference of human single-cell RNAseq data[27]. Our cells mapped onto the corresponding

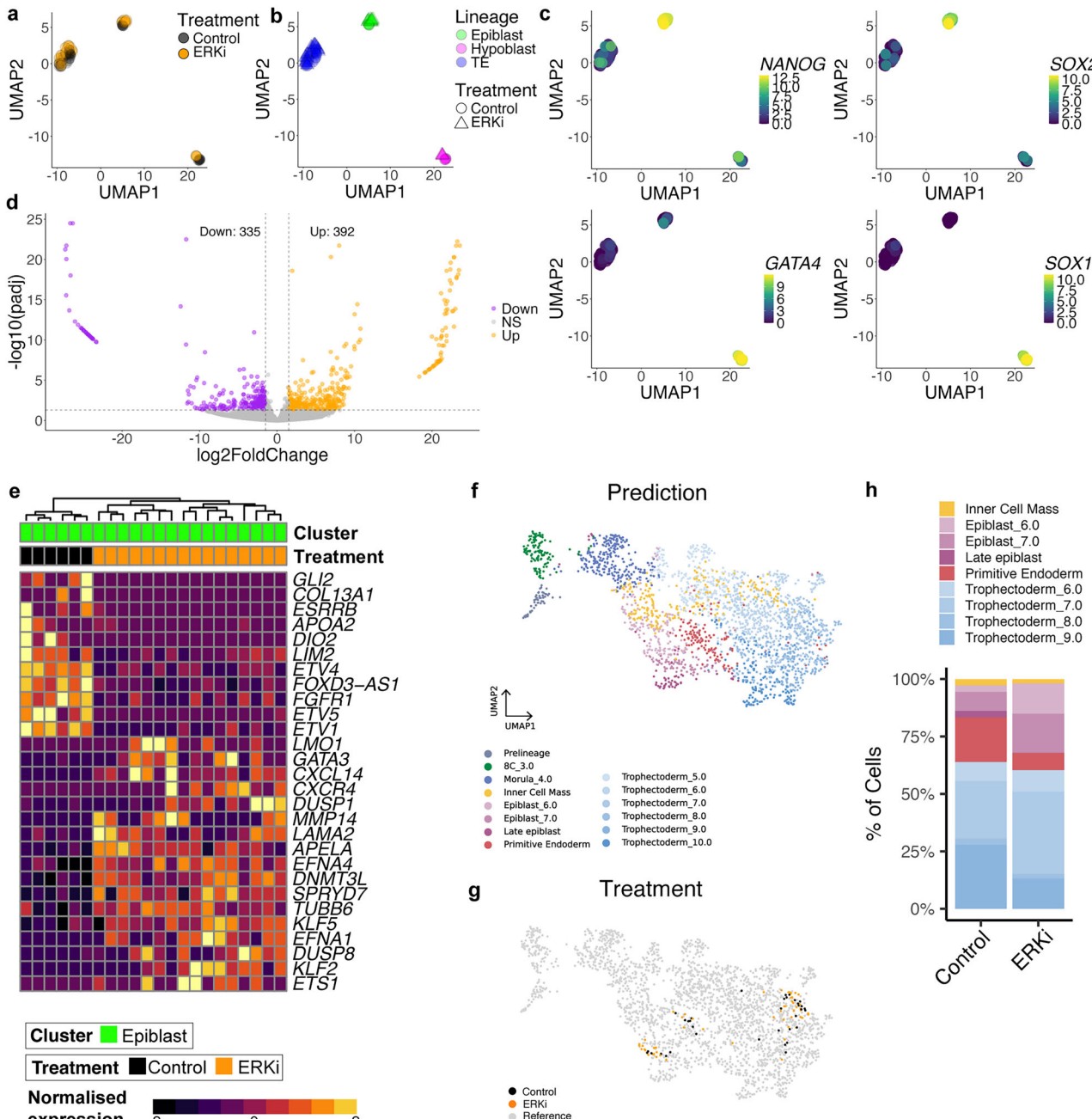

**Fig. 3 | ERK inhibition alters lineage specification and maintains naïve pluripotency characteristics in human embryos. a** UMAP of single cells from Day 6.5 human embryos cultured in ERKi (orange) or control medium (black). **b** UMAP coloured according to lineage identity. **c** UMAP coloured by expression of epiblast or hypoblast marker genes. Scales show log transformed normalised expression values. **d** Volcano plot with genes significantly differentially expressed between ERKi treated and control cells within the epiblast cluster in (**a**). Significantly genes (DESeq2 padj <0.05 and log2FC > 1.5) upregulated (orange) or downregulated (purple) in ERKi vs control. Adjusted $p$-values were calculated using the two-sided Benjamini–Hochberg method to control for FDR across multiple tests.

**e** Hierarchical clustering and heatmap of selected significantly differentially expressed genes between ERKi treated and control cells within the epiblast cluster. Normalized expression shown as z-score. **f, g** UMAP integrating this study with a compiled human embryo reference dataset[28], coloured according to https://doi. org/10.1101/2024.02.01.578414 (**f**) lineage prediction using a deep learning-model and (**g**) treatment group from this study Control (black) and ERKi (orange). Reference data set is under control conditions. **h** Stacked bar chart showing lineage prediction of cells from this study using a deep learning-model in control and ERKi conditions. Source data are available on Github.

branches of the human reference dimensionality reduction plot based on their lineage annotations (Supplementary Data Fig. 3g–i). The prediction tool[27] showed a high level of accuracy (97%) between our manual annotation and the reference.

To further probe the cell identities and the embryonic stages they correspond to in an unbiased fashion, we applied our dataset to our recently developed deep learning-model[28] (Fig. 3f, g). Prediction using

the deep-learning model and integration with reference human single-cell RNAseq data showed a high level of accuracy (96%) between our manual lineage annotation and the reference. The predictions confirmed the proportion of epiblast cells vs hypoblast cells was enriched in the ERKi treatment, indicating a cell fate switch in the ICM (Fig. 3h). In both the epiblast and trophectoderm lineage, the model predicts there is a higher proportion of cells corresponding to earlier

embryonic stages in the ERKi treatment. For example, ERKi epiblast more closely resemble more immature (or naïve) epiblast at Day 6 of human development, compared to control epiblast cells, that contain a greater proportion of more mature Day 7 and late epiblast cells.

Together, these findings demonstrate that ERK inhibition skews ICM fate towards a naïve-pluripotent epiblast identity, and disrupts normal developmental progression, highlighting a critical role for ERK in human blastocyst lineage commitment.

### Comparative single-cell transcriptomics reveals conserved ERK function in hypoblast specification but divergence in pluripotent epiblast lineage maintenance

We next sought to understand if the effect of ERKi on cell fate may be similar between mammalian species, therefore, we performed single-cell RNA-seq analysis after immunosurgery on ERKi and untreated control mouse embryos, over similar time scale to the human, from early- to late- blastocyst stage (E3.25 + 24 h). We analysed 26 ICM cells from 4 ERKi treated embryos, and 25 ICM cells from 5 control embryos. Dimensionality reduction followed by marker analysis clustered the cells into two distinct populations, which were assigned as epiblast (*Nanog, Sox2, Fgf4*) or hypoblast (*Gata6, Gata4, Sox7*) (Supplementary Data Fig. 4a–c). Only epiblast, and no hypoblast cells were present in the ERKi treatment, indicating a total shift in cell fate within the ICM (Supplementary Data Fig. 4d).

We next performed differential gene expression on epiblast cells in the two conditions (Supplementary Data Fig. 4e, Supplementary Data 2). The top enriched gene sets; mTOR signalling, G2/M checkpoint, Myc and E2F2 targets, were downregulated in ERKi epiblast cells (Supplementary Data Fig. 4f). These gene sets are involved in the regulation of the cell cycle and diapause[29,30], and suggest the ERKi epiblast may be in a dormant pluripotent state. For individual differentially expressed genes, those associated with RTK (*Dusp14, Dusp9, Btk, Egr1*), STAT3 (*Junb, Fos*) signalling and pluripotency (*Tfap2c, Foxd3, Dnmt3b, Lin28a*) were misexpressed (Supplementary Data Fig. 4g). Only a few differentially expressed genes (4 / 224) were misregulated in a concordant manner with their orthologous genes in ERKi human embryos (i.e. the same direction of expression). Furthermore, although mTOR signalling pathway gene set was enriched for both species, in mouse it was downregulated, and human was upregulated. Together these data indicate divergent functions of ERK in the epiblast between the species.

We next integrated our dataset with published mouse single-cell RNAseq reference datasets[28]. After dimensionality reduction analysis, the cells from our dataset mapped most closely to E4.5 epiblast (EPI) and E4.5 primitive endoderm (i.e hypoblast, PrE) reference cells on a force directed graph, corresponding to the embryonic day at collection (Supplementary Data Fig. 4h, i). Using our deep learning model to classify cell types, there was a good level of concordance (88%) with our manual annotation (Supplementary Data Fig. 4j). Hypoblast cells were only identified in control conditions, with some E4.5 EPI ERKi reclassified as E3.5-ICM, suggesting that ERKi epiblast may be an earlier stage of development compared with controls (Supplementary Data Fig. 4k).

Together, these findings indicate that while ERK inhibition biases ICM fate toward epiblast in both species, its downstream effect in lineage progression differs, suggesting evolutionary divergence in ERK-dependent epiblast networks.

### Suppression of ERK signalling promotes pluripotent epiblast identity in human

To functionally test the pluripotent potential of the epiblast following ERKi treatment of human embryos, we aimed to derive naïve hESC lines. Previously, naïve hESCs, the in vitro stem cell counterpart of the pre-implantation epiblast, were derived directly from isolated ICM cells in conditions containing a titrated concentration of GSK3, MEK and PKC

inhibitors together with LIF (t2iLGö) medium[31–33]. We initially tested to see if we could directly derive naïve hESCs from human blastocysts in the further optimized medium where the GSK3 inhibitor has been replaced with XAV939, a tankyrase inhibitor and Wnt pathway antagonist (PXGL medium)[34]. We plated Day 6 human blastocysts, intact or following mural trophectoderm laser dissection. We were able to derive 5 stable naïve hESC cell lines from 8 blastocysts in both intact and dissected conditions (Supplementary Data Fig. 5a–c, (i) demonstrating efficient naïve hESC derivation in PXGL medium.

Immunofluorescence analysis of Day 5 embryos following 36 h ERKi treatment showed that human epiblast not only retained protein expression of SOX2, but also expressed KLF17, a molecular marker associated with naïve pluripotency, giving confidence that we could derive naïve pluripotent lines from these later Day 6.5 embryos (Fig. 4a). After 36 h in either ERKi or control culture, we plated whole intact Day 6.5 blastocysts in naïve PXGL medium to derive naïve hESCs (Supplementary Data Fig. 5a, (ii)). After ICM outgrowth and passaging, we were able to establish 2/8 naïve hESC lines from control embryos and 5/9 naïve hESC lines from the ERKi treated embryos (Fig. 4b). These lines had characteristic compact and domed naïve ESC morphology (Fig. 4c), and immunofluorescence analysis showed they expressed core pluripotency markers OCT4 and SOX2, and naïve pluripotency markers KLF4, KLF17, SUSD2, and TFAP2C[35–38] (Fig. 4d,e), and were karyotypically normal (Supplementary Data Fig. 5d).

We next sought to determine if the MEK inhibitor PD0325901 ("P" in PXGL media) in naïve pluripotent stem cell derivation and maintenance could be replaced with our ERKi Ulixertinib ("U" in UXGL media). Following 36 h of either ERKi or DMSO treatment we plated the laser dissected blastocysts in UXGL media (Supplementary Data Fig. 5a, (iii). Following ICM outgrowth and subsequent passaging, we established 1/4 naïve hESC from control embryos and 2/4 naïve hESC lines from ERKi treated embryos (Supplementary Data Fig. 5b, (iii)). As above, the UXGL hESC lines had characteristic domed shaped morphology and expressed SOX2, NANOG and OCT4 in addition to naïve pluripotency markers KLF4, KLF17, and DPPA5 (Supplementary Data Fig. 5c, e),

The transcriptional profiles of hESC derived in PXGL or UXGL media, whether following ERKi or control embryo treatment, were more similar to previously established naïve hESC than primed hESC, as determined by bulk RNA-seq of these lines[15,32–34,36,39] (Fig. 4f). Moreover, the PXGL and UXGL hESC were similar to previously derived naïve hESC in their high expression of naïve-pluripotency markers, low expression of primed-pluripotency markers, and equivalent level of core pluripotency markers compared to primed hESC (Fig. 4g). PXGL and UXGL hESC derived following ERKi or control treatment also resembled previously established naïve hESC in transcriptional similarity to human pre-implantation epiblast[17–19] cells compared to primed hESC (Fig. 4h).

Global DNA hypomethylation is characteristic of mouse and human ICM, and naïve PSC. We performed whole-genome bisulphite sequencing to determine the methylome of our newly derived ESC lines. We performed principal component (PC) analysis to compare the methylomes with previously published datasets[40–42]. PC1 explained 85% of the variance between samples, with published primed hESC clustering away from human blastocyst, as well as published and our new naïve hESC lines (Fig. 4i). Primed cells showed uniformly high levels of global DNA methylation (85 95%), whereas naïve lines showed hypomethylation (35 45%), and our control and ERKi PXGL and UXGL lines showed similarly low levels of methylation (45–55%) (Fig. 4j, and Supplemental Data Fig. 4f). Imprinted differently methylated regions were also demethylated in our control and ERKi PXGL and UXGL lines, in line with published naïve hESC (Supplemental Data Fig. 4g)

In conclusion, ERKi embryos retain pluripotency markers throughout their ICM and give rise to ground-state pluripotent stem cells, demonstrating their naïve potential.

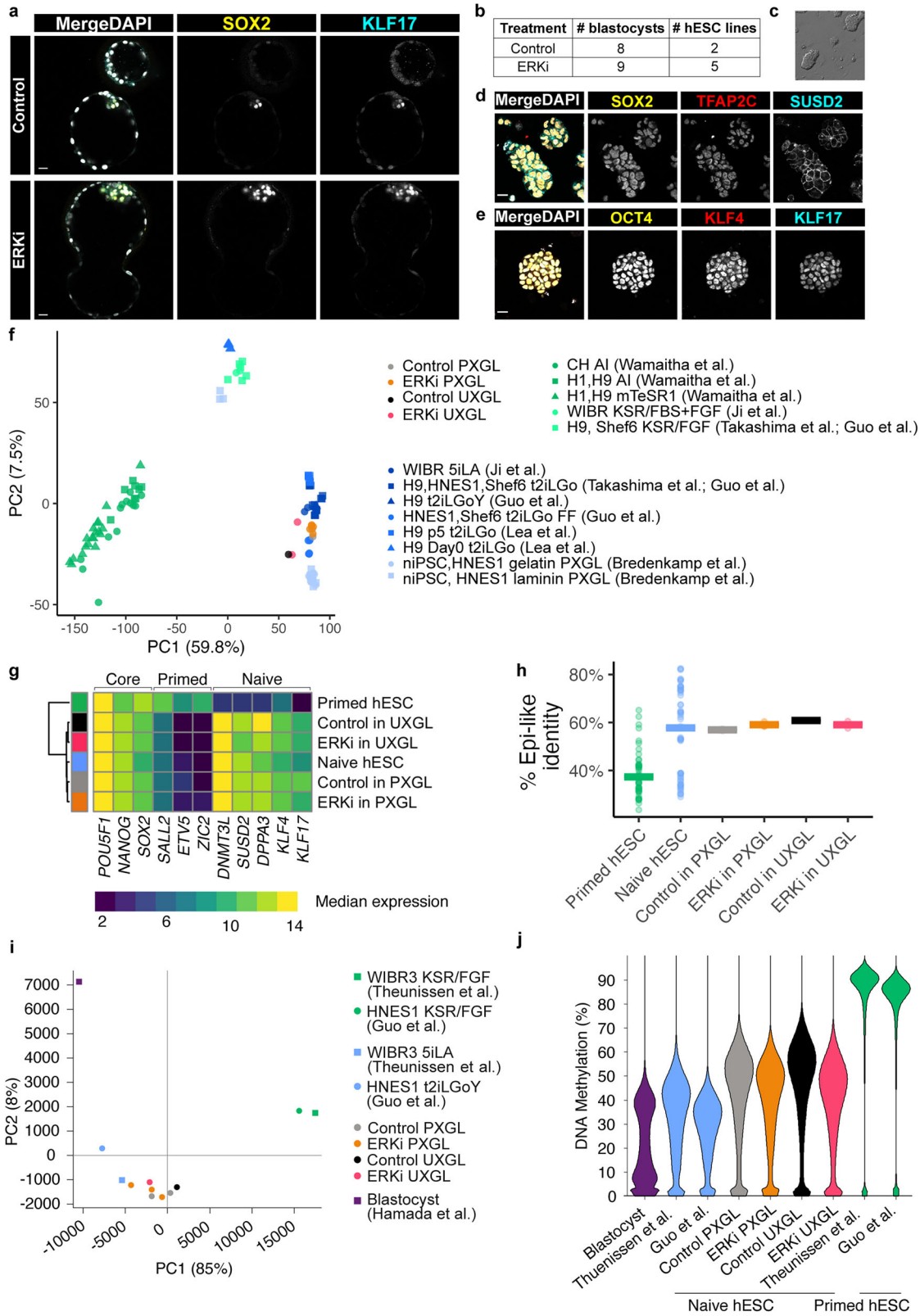

## Discussion

Our data demonstrates that FGF4 is sufficient to drive human hypoblast specification in a dose dependent manner. Divergence of epiblast and hypoblast markers between FGF4 expressing and non-expressing cells in the human ICM further supports FGF/ERK governing lineage segregation[43]. However, it is still unclear which ligands drive this signalling activity in vivo. Further work is needed in vivo to confirm

endogenous human FGF4 function in hypoblast specification, and whether other RTKs such as IGFs[15], or parallel pathways such as WNTs, as has been suggest in marmoset monkeys[44], may additionally be involved.

The necessity for FGF/ERK in human hypoblast specification has also been suggested by stem cell models, where conversion of naïve hESC to hypoblast-like cells is dependent on FGF signalling[16,45].

**Fig. 4 | Suppression of ERK signalling promotes pluripotent epiblast identity.** **a** Confocal images of Day 6.5 human embryos immunofluorescently labelled for SOX2 and KLF17 (epiblast) and stained for nuclear DAPI. Scale bars 20 μm. **b** Table showing outcome of naïve hESC derivation from Control and ERKi cultured embryos in PXGL medium (Control PXGL; ERKi PXGL). **c** Phase-contrast image of an ERKi PXGL naïve hESC line. **d**, **e** Confocal images of an ERKi PXGL naïve hESC line, immunofluorescently labelled for naïve (TFAP2C, SUSD2, KLF4, KLF17) and core (SOX2, OCT4) pluripotency markers and stained for nuclear DAPI. Scale bars 20 μm. Representative images Control $n = 2$, ERKi $n = 5$. **f** Principal component analysis following RNA-seq of hESC derived in this study and previously published primed and naïve hESC lines. Per-gene variance was modelled, and significantly variable genes (DESeq2 padj < 0.05) were used in the loading. Adjusted $p$-values were calculated using the two-sided Benjamini–Hochberg method to control for FDR across multiple tests. **g** Heatmap showing the normalised median expression of selected pluripotency genes in previously published hESC and from this study. **h** Dot plots showing the relative expression of human pre-implantation[17–19] epiblast-enriched genes (% Epi-like identity) for individual hESC lines. Horizontal bars indicate median gene expression in each group. RNA-seq analysis (f-h) of published primed ($n = 48$) or naïve ($n = 55$) hESC[15,32–34,36,39] and hESC derived in this study (Control PXGL $n = 2$, ERKi PXGL $n = 5$, Control UXGL $n = 1$, ERKi UXGL $n = 2$). **i** Principal components analysis of DNA methylome in published blastocyst, primed hESC and naïve hESC, and hESC derived in this study. **j** Beanplots showing distribution of DNA methylation for published blastocyst, primed hESC, naive hESC, and hESC derived in this study. Whole-genome bisulphite sequencing analysis (**i**, **j**) of published blastocyst ($n = 1$), primed hESC ($n = 2$), and naïve hESC ($n = 2$)[40–42] and hESC derived in this study: Control in PXGL ($n = 2$), ERKi in PXGL ($n = 3$), Control in UXGL ($n = 1$) and ERKi in UXGL ($n = 1$). DNA Methylation is quantified using autosomal 100-CpG windows. Source data are available on Github.

However, attempts at blocking the FGF/MEK/ERK pathway using small molecule inhibitors against MEK or FGFR produced varying outcomes in mammalian embryos, and might argue for intrinsic species-specific differences in the molecular mechanism of ICM cell fate decisions. For example, while mouse, rat and rabbit embryos all lost hypoblast cells upon MEKi, only mouse and rat have an accompanying expansion of epiblast cells[2,5,14]. By contrast, in pig, cow, and marmoset embryos MEKi did not completely block hypoblast specification, but the numbers of cells were reduced[3,4,44]. Most surprisingly, in human embryos, MEKi and/or FGFRi did not affect hypoblast formation in a significant manner[3,14]. Potential confounding experimental factors include FGFRi or MEKi concentrations, treatment duration, volume of mineral oil[46], diffusion across the zona, and cross-talk from other pathways[15], which may all impact treatment efficacy in sufficiently inhibiting ERK activity. A study published while our manuscript was under review used human stem cell-based embryo models and exogenous FGF2 or FGFRi treatment of human embryos[47], and corroborates the findings we present here, that FGF lies upstream of ERK in executing hypoblast specification. Together, our studies overturn previous conclusions about the role of FGF/ERK signalling in human embryo cell fate decisions[48].

FGF/ERK signalling plays a critical role in epiblast development and naive pluripotency. We find that in human embryos, inhibiting ERK signalling leads to expansion of epiblast marker expression throughout the ICM, with high levels of pluripotency markers and competence for naïve hESCs derivation. The upregulation of naïve pluripotency markers (e.g. *KLF2, KLF5, DNMT3L*) in embryos parallels findings in hESC where ERK inhibition can be used to maintain naïve pluripotency[49]. We demonstrate that ERK inhibition can substitute for MEK inhibition in the generation of stem cell lines from embryos, thereby establishing an alternative method for naïve hESC derivation. Notably, in human embryos we find a decrease in critical downstream FGF targets, *ETV4* and *ETV5* which are required for epiblast maturation and pluripotency progression in mouse embryos[50]. Together suddests that ERK signalling in human is required for the maturation of the epiblast and transition out of a naïve pluripotent state.

Our comparison across mouse, rat, cow and human embryos demonstrates commonalities in ERK regulating the second cell fate decision. While ERK inhibition biases ICM fate to epiblast, its effect on lineage progression diverge—maintating naïve pluripotency in human and inducing dormant pluripotent state in mouse. Underscoring the importance of comparative studies on understanding of the molecular mechanisms governing early mammalian development.

Several models have been proposed for specification of the three lineages in human preimplantation development. Our data supports a two-step sequential model of lineage segregation; an intial trophectoderm versus ICM specification followed by segregation of the ICM to epiblast and hypoblast cells. This agrees with our functional data revealing the importance of cell polarity and Hippo signalling activity in regulating ICM versus trophectoderm in the first cell fate decision[51,52]. Our data is also consistent with recent re-analysis of single-cell RNA-seq data together with protein analysis characterising ICM progenitor cells as they commit to epiblast and hypoblast in the second cell fate decision[26,53]. Altogether, we show that FGF/ERK activity can alter this balance between epiblast and hypoblast, and that the level of activity in a common ICM progenitor determines the balance between these two lineages.

## Methods
This study complies with all relevant ethical regulations. This study was approved by the UK Human Fertilisation and Embryology Authority (HFEA): research licence numbers R0162, R0397, R0401 and R0152 and independently reviewed by the Health Research Authority's Research Ethics Committee IRAS projects 308099, 252286 and 272218.

### Ethics statement
The process of licence approval entailed independent peer review along with consideration by the HFEA Licence and Executive Committees and the Research Ethics Committee. Our research is compliant with the HFEA Code of Practice and has undergone multiple inspections by the HFEA since the licence was granted.

Informed consent was obtained from all couples that donated spare embryos following infertility treatment. Before giving consent, people donating embryos were provided with all the necessary information about the research project, an opportunity to receive counselling and the conditions that apply within the licence. Donors were informed that embryos used in the experiments would be stopped before 14 days post-fertilization and that subsequent biochemical and genetic studies would be performed. Informed consent was also obtained from donors for all the results of these studies to be published in scientific journals. No financial inducements were offered for donation. Consent was obtained for creation and culture of embryonic stem cell lines from these embryos and deposition of cell lines in the UK Stem Cell Bank. Embryos surplus to the patient's IVF treatment were donated cryopreserved and were transferred to the University of Cambridge and Francis Crick Institute, where they were thawed and used in the research project.

### Human embryo thaw
Human embryos were thawed using vit kit-Thaw (Fujifilm) or Embryo Thawing Media (Kitazato, VT602) for vitrified embryos, or BlastThaw™ Kit (CooperSurgical) for slow frozen embryos according to the manufacturer's instructions.

### Human embryo culture
Human blastocysts were cultured in pre-equilibrated Global Media (Cooper Surgical) supplemented with 10% Human Serum Albumin (HSA, Cooper Surgical) in Embryo+ slide dishes (Vitrolife) overlaid with mineral oil (Cooper Surgical). Embryos for the second batch of single-cell RNAseq were alternatively cultured in GT-L culture media (Vitrolife). Embryos were incubated at 37 °C 5.5% $CO_2$ in an EmbryoScope

time-lapse imaging system (Vitrolife). For treatments, immediately after thawing embryos were incubated in Global media, 10% HSA supplemented with; 250 ng/ml FGF 4 (R&D) and 250 ng/ml Heparin (Sigma); 500 ng/ml FGF4 and 500 ng/ml Heparin; 750 ng/ml FGF4 and 750 ng/ml Heparin; 5 μM Ulixertinib (Cambridge Bioscience); 0.1% DMSO. Day 5 embryos were treated with cytokines and inhibitors for either 36 or 24 h.

## Embryo immunofluorescence staining

Human, mouse and rat embryos were washed briefly in PBS−/− + 1% HSA, followed by fixation in 4% PFA (Fisher Scientific) on ice for 1 h. Embryos were then washed three times in PBS-/- + 0.1% Triton X-100 (PBX; Sigma), and permeabilized in PBS-/- + 0.5% TritonX-100 for 20 min. Embryos were washed briefly in PBX, then incubated in blocking solution: 10% Donkey serum (Jackson ImmunoResearch, 017-000-121) in PBX, for 1 h on rotating shaker. Embryos were incubated with primary antibodies (Supplementary Table 1) diluted in blocking solution overnight at 4 °C on rotating shaker. The next day, embryos were washed three time in PBX, and incubated in blocking solution for 1 h on rotating shaker, before incubating with secondary antibodies diluted in blocking solution for 1 h on rotating shaker. Finally, embryos were washed four times in PBX, and a final wash PBS-/- + 0.1% Triton + 3.33% Vectashield with DAPI (Vector Laboratories, H-1200). For staining of pERK, embryos were stained as above, with the following modifications: 1 x PhosSTOP (Roche, 4906845001) was included in all solutions up to and including the primary antibody incubation, embryos were fixed in 8% PFA + 1x PhosSTOP at room temperature for 10 min on a rotating shaker. Primary and secondary antibodies used in this study are outlined in Supplementary Table 1.

## Embryo immunofluorescence imaging

Fixed samples were imaged on a Leica SP8 scanning confocal microscope or a Zeiss LSM880. Embryos were mounted in microdroplets of PBS + 0.025% Tween20 + 1.5% Vectashield, on glass bottomed 35 mm dishes (Maktek, P35G-1.5-14-C) coated with mineral oil. Embryos were imaged along the entire z-axis with a 1 μm step using a glycerol immersion HC PLAN APO CS2 63 × 1.30 NA objective (Leica), oil immersion HC PLAN APO CS2 40 × 1.30 NA objective (Leica) or a 40x or 20x objective (Zeiss). Imaging parameters were kept consistent across experiments.

## Image processing and quantification

Raw images were processed in ImageJ. Nuclear segmentation for quantification of nuclear and cytoplasmic fluorescence was carried out using Stardist and CellProfiler as described previously[36]. Correction of segmentation errors and classification of inner ICM vs outer TE cells was performed manually, and trophectoderm marker (GATA3) used as an additional classifier for FGF and ERKi experiments. Adjustments for fluorescence decay along the z-axis was performed by linear regression followed by empirical Bayes method[54]. Hierarchical clustering was used to classify ICM cell lineage based on adjusted nuclear fluorescence intensities of the hypoblast marker (GATA4 or OTX2) and epiblast marker (NANOG or SOX2) for FGF, ERKi and pERK experiments. To quantify the nuclear versus cytoplasmic levels of pERK, nuclear segmentation was expanded by 4 pixles to generate a cytoplasmic ring around the nucleus and the ratio of nuclear or cytoplasmic expression was determined. The Github repository includes further step-by-step details of the CellProfiler pipelines used.

## Embryo dissection and dissociation

Human and mouse embryos were subject to immunosurgery to remove outer trophectoderm cells and isolate the inner cell mass (ICM), as reported previously[31,55]. The zona pellucida was removed by washing embryos through drops of Acidic Tyrode's solution (Merck) pre-warmed to 37 °C and overlaid with mineral oil. Embryos were then washed briefly and incubated at 37 °C for 30 min in 20% anti-mouse serum antibody (Sigma) or 20% anti-human serum antibody (Sigma), depending on the species, in Global media + 10% HSA. Embryos were then washed briefly in Global Media + 10% HSA, before incubation with 20% Guinea Pig Serum Complement (Merck) at 37 °C for 10–15 min until trophectoderm cell began to lyse. Embryos were then moved to a drop of medium and triturated with 100 μm STRIPPER tip (Cooper Surgical) to remove lysed trophectoderm cells and to isolate the inner cell mass. Human embryos for the second batch of single-cell RNA-seq were subjected to laser dissection (Saturn 5 Laser) to remove the mural trophectoderm, as described previously[17].

Isolated ICMs were then briefly washed, then incubated at 37 °C for 3 min in 0.5% Trypsin, 1 mM EDTA in PBS (Fisher Scientific). Then, ICMs were transferred to 4% BSA, 0.5 mM EDTA in PBS for fine manual dissociation into single cells with pulled (Sutter Instruments) glass capillaries (World Precision Instruments). Cells were then picked manually with pulled glass capillaries for downstream single-cell RNA-seq.

## Single cell RNA-seq

Single-cell cDNA synthesis was performed using SMART-Seq v4 Ultra Low Input RNA Kit for Sequencing (Takara) according to the manufacturer's protocol with some modifications as previously described[17]. Briefly, single-cells were manually picked after dissociation and snap frozen on dry ice in 5 μl 10x Reaction Buffer in low-bind 0.2 ml PCR tubes and stored at −80 °C until processing. Samples were subject to first strand cDNA synthesis, then cDNA was amplified by LD-PCR for 23 cycles. Amplified cDNA was purified on AMPure XP beads, using a 96-well magnetic stand (ThermoFisher), and eluted in 17 μl of elution buffer. Single-cell RNA-seq libraries were prepared using a Nextera XT DNA Library Preparation Kit (Illumina) according to the manufacturer's protocol. Sequencing was carried out as paired-end 50 bp read on Novaseq 6000 to a depth of ~12 million reads per cell. Modifications for the second batch of human single-cell RNA-seq include the use of SMART-Seq mRNA (Takara), snap freezing in 10.5 μl 1x reaction buffer, and performing LD-PCR for 18 cycles. The libraries were prepared using the SMART-Seq mRNA LP kit (Takara) according to the manufacturer's protocol. Sequencing was carried out as paired-end 150 bp read on an AVITI sequencer.

## Single cell RNA-seq processing and analysis

Raw sequencing was processed according to a modified nf-core/scrnaseq pipeline[56] as described previously[28]. In brief, reads were trimmed to remove adaptor sequences and aligned to the human GRCh38 or mouse GRCm38 reference genomes using STAR aligner. Transcript abundances, using the Ensembl gene annotations (versions 110 and 102 for human and mouse, respectively) were then estimated using STAR quantmode, converted into a count matrix and further processed using scanpy. No computational doublet detection was performed. For mouse blastocyst sequencing, samples with >15% of reads aligning to the mitochondrial genome were excluded. In the case of human blastocysts, we excluded samples in which >30% of total reads aligning to the mitochondrial genome and fewer than 8000 unique genes were detected.

For assigning cell identities, mitochondrial genes, pseudogenes, ribosomal genes and genes with an average count <1 across all samples were removed prior to normalization. After gene-level filtering, normalization, scaling, and log-transformation were performed and per-gene variance was modelled with scran[57] and the top 10% most variable genes were used for principal component analysis (PCA) and inputted for UMAP analysis for clustering to asses lineage identity. In mouse, only 1 control trophectoderm was identified, indicating successful immunosurgery, and excluded from downstream analysis. Differential expression between the ERKi and control samples was determine using DESeq2 using the default standard Wald test, and significantly

differentially expressed genes were identified as Padj <0.05 and absolute Log2FC > 1.5. Gene set enrichment analysis (GSEA) was permed on the DESeq2 output, using Human and Mouse Molecular Signatures Database (MSigDB) Collection H: Hallmarks v2024.1. Integration with published human single-cell RNA-seq reference was performed using the Early Embryogenesis Projection Tool (v2.1.2)[27] https://petropoulos-lanner-labs.clintec.ki.se/shinys/app/ShinyEmbryoProjP.

For integration to mouse and human reference datasets[28], gene counts were first normalized to both mean gene length and 10,000 reads and the top 3,000 highly variable genes were extracted. Cell identity predictions were assigned using a deep learning model trained on single cell RNA-seq of early mouse and human embryogenesis. We used recently published preimplantation models[28] and re-trained them with scvi-tools v1.1.5. The mouse reference model was adjusted by setting cells labelled as E3.75 ICM to Unknown and letting the model to assign either an epiblast or primitive endoderm identity. Gene length and sequencing depth normalized transcriptomes were assigned using scANVI query; the uncertainty of assignment or entropy was defined as 1-highest prediction probability.

## Human naïve ESC derivation

Human Day 5 embryos were cultured for 36 h in ERKi or in DMSO control medium or thawed on Day 6 and cultured for 2 h in Global medium. Embryos that had not yet hatched had the zona pellucida removed by a brief wash in Acidic Tyrode's pre-warmed at 37 °C. Whole intact embryos, or laser dissected embryos to remove mural trophectoderm, were then plated on MEF coated 4-well plates (Nunc) in pre-equilibrated PXGL[34] or UXGL medium. PXGL or UXGL medium was made from N2B27 medium (Takara, Y40002), supplemented with either 1 μM PD0325901 ("P", Cambridge Bioscience; 13034-1mg-CAY) or 5 μM Ulixertinib ("U", Cambridge Bioscience) plus: 2 μM XAV939 ("X", Cambridge bioscience, CAY13596-1mg), 2 μM Gö6983 ("G" Bio-Techne; 2285/1), 10 ng/ml hLIF ("L", PeproTech; 300-05), and Pen/Strep (Gibco). Embryo outgrowths were left undisturbed for 48 h at 5% $O_2$ 5%$CO_2$ 37 °C, then fed with pre-equilibrated half medium changes every 2 days. After 7−10 days, outgrowths were manually picked under a dissecting microscope. Outgrowths were dissociated by washing, and then incubated in microdrops of Accutase under mineral-oil for 5 min at 37 °C before trituration into single cells with 290 μm and 100 μm STRIPPER tips (CooperSurgical). Dissociated single cells were then washed in PXGL or UXGL medium and plated onto MEF feeder plates with PXGL or UXGL medium supplemented with ROCK inhibitor 10 μM Y27632 (Tocris), and then re-fed with PXGL or UXGL medium every 2 days. Clones were passaged manually every 5−7 days until sufficient numbers for bulk passaging, initially in a 1:1 split ratio, then 1:2−1:3 split ratio every 3−5 days.

## Human naïve ESC culture

Human naïve cells were cultured in PXGL medium[34]: N2B27 medium (Takara, Y40002), supplemented with 1 uM PD0325901 (Cambridge Bioscience; 13034-1 mg-CAY), 2uM XAV939 (Cambridge bioscience, CAY13596-1mg), 2 μM Gö6983 (Bio-Techne; 2285/1), 10 ng/ml hLIF (PeproTech; 300-05), and Pen/Strep (Gibco), on MEF feeder layer (~1 × 105 cells / cm²) and incubated under hypoxic conditions 5%$O_2$ 5% $CO_2$ 37 °C in a humid incubator. Culture medium was replaced every other day, and cells were passaged every 4−5 days with a split ratio of 1:3−1:5. Cells were passaged by washing with PBS, followed by incubation with Accutase (Thermo, A1110501) at 37 °C for 5-10 min with gentle pipetting and monitoring to confirm single-cell dissociations. Cells were washed and pelleted, before resuspension and culture in PXGL+ROCKi medium for 24 h on a fresh MEF feeder plate. After 24 h, the medium was replaced with fresh PXGL (without ROCKi). Alternatively, the 1 μM PD0325901 above was replaced with 5 μM Ulixertinib (Cambridge Bioscience) to generate UXGL medium.

## Human ESC immunofluorescence and imaging

Primed hESC (seeded at 1:10) and naïve hESC (seeded at $5 × 10^4$ cells per well) were grown in μ-Slide 8 well high chambered coverslip (Ibidi, 80806). Prior to staining cells were washed briefly in PBS, before fixation in 4% PFA at room temperature for 10 min. Cells were washed three times in PBS + 0.1%Triton-X (PBX) before 20 min permeabilization in 0.5% Triton X-100 on a rotating shaker. Cells were then washed, incubated in blocking solution (10% Donkey serum in PBX) for 1 hr at room temperature. Then, cells were incubated overnight in primary antibody in blocking solution at 4 degrees. Cells were washed three times in PBX, before a second incubation in block solution for 1 hr, followed by incubation with secondary antibodies. Antibodies outlined in Supplementary Table 1. Samples were imaged in PBS on a Leica SP8 scanning confocal microscope using a dry HC PLAN APO CS2 20 × 0.75 NA objective (Leica)

## Human ESC RNA-seq

Stable naïve hESC lines were used for the bulk RNA-seq experiments. Human naïve ESCs grown in PXGL were dissociated to single cells. $6 × 10^5$ cells were resuspended in 600 μl RLT buffer (Qiagen) and snap frozen on dry ice for later processing. RNA was isolated using the RNAeasy Mini Kit (Qiagen, 74104), QIAShredder DNase digest (Qiagen, 79654), automated on a QIAcube (Qiagen) according to the manufacturer's instructions. The RNA library prep was performed using the mRNA polyA (NEB) kit, before paired-end 100 bp sequencing on a NovaSeq2 (depth of 25 M reads).

RNA from Human naïve ESCs, ~1-5 × 105 cells, grown in UXGL were harvested using TRIzol reagent (ThermoFisher Scientific), according to the manufacturer's instructions. Residual TRIzol was removed from the RNA samples by chloroform purification, followed by isopropanol precipitation. The RNA samples were treated with TURBO DNase (ThermoFisher Scientific) in the presence of RNaseOUT recombinant ribonuclease inhibitor (ThermoFisher Scientific). DNase was removed from the RNA samples by adding Phenol:Chloroform:Isoamyl Alcohol (25:24:1, v/v) pH 4.5, followed by ethanol precipitation. PolyA selected RNA was sequenced paired-end 150 bp, on Illumina NovaSeq (depth of 30 M reads).

## Human ESC RNA-seq analysis

Reads were assessed for quality with FastQC and trimmed with Trim-Galore 0.5.0. Reads were pseudoaligned to GRCh38 using Salmon 0.11.3. Mitochondrial genes, pseudogenes, ribosomal genes, and genes not detected in any condition were removed. Technical duplicates of hESC UXGL lines were summed. Samples were normalised using DESeq2 and invariant genes were removed. Any single-cells present were subject to imputation using DRImpute. Per-gene variance was modelled with scran 1.26.2 and variable genes FDR < 0.05 were used as input into principal component analysis. Similarity to an epiblast identity was assessed using DeconRNASeq[58], based on a previous publication[34], using as a reference set high confidence epiblast cells that were obtained from previous work[59].

## Bisulphite-sequencing

Low input bisulphite-sequencing libraries were generated using ~5000 cells, as previously described, using a post-bisulphite adaptor tagging (PBAT) method[60]. In brief, cells suspended in PBS were lysed in 10 mM Tris buffer with 0.5% SDS and proteinase K at 37 °C for 1 h. Bisulphite conversion was done using the EZ DNA Methylation Direct Kit (Zymo Research), as per the manufacturers protocol. First and then second strand synthesis was performed on eluted DNA using Klenow Fragment (3'- > 5' exo-) enzyme (New England Biolabs) and customised 9 bp random sequence containing biotin-conjugated adaptors. Using Phusion High-Fidelity DNA polymerase (New England Biolabs), libraries were amplified for 10 cycles in a 50 μL reaction. Libraries were purified using SPRI beads and then quantified using the High DNA Sensitivity

Bioanalyzer 2500 (Agilent) and Illumina library quantification kit (KAPA). Samples were multiplexed and sequenced using 75 bp paired-end mode on the AVITI sequencer.

## Bisulphite-sequencing data processing

PBAT data had the first 9 bp of both read 1 and read 2 removed to reduce biases arising from the 9 N oligo pull-down reaction (Trim Galore options: -clip_r1 9 -clip_r2 0 -paired). PBAT-sequencing libraries were quality and adaptor trimmed with Trim Galore v0.6.6 with -clip parameters. Libraries were deduplicated and methylation calls extracted using PBAT mode in Bismark v0.16.3[61] with paired-end alignment to the human GRCh38 genome assembly.

## DNA methylation analysis

In addition to the datasets published in this study, DNA methylation from blastocysts (JGAS00000000006)[42] and naïve and primed hESCs (GSE75868 and E-MTAB-4462)[32,41] is also included. CpG methylation files were loaded into SeqMonk Mapped Sequence Data Analyzer software (v 1.48.0) for visualisation and analysis. Autosomal 100-CpG running windows ($N = 242,555$) were defined based on the CpGs assayed in DMSO_PXGL, ULIX_PXGL, DMSO_UXGL, ULIX_UXGL replicate sets. DNA methylation values were quantitated using the bisulphite-sequencing pipeline with a minimum coverage of 10 CpGs in each sample. Classic imprinted DMRs analysed in this study were previously identified[62]. For each dataset, DNA methylation values were quantitated using the bisulphite sequencing pipeline with a minimum coverage of 5 CpGs in each sample and combined as mean.

## Statistics and reproducibility

Sample size was determined based on our previous experience and experience from other groups' work. Arrested embryos were excluded from the analyses. The experiments were randomized and where possible embryos donated from the same donor were split between control and experimental groups. The investigators were not blinded to allocation during experiments and outcome assessment.

The number of cells or embryos analysed ($n$), statistical tests and $p$-values are all stated in each figure or figure legend. The tests performed in this paper are unpaired two-tailed Student's $t$-test. For multiple comparison of RNAseq analysis, false discovery rates were controlled with the Benjamini–Hochberg method. Data are represented as mean +/- s.d. Unless otherwise noted, each experiment was performed at least three times.

## Reporting summary

Further information on research design is available in the Nature Portfolio Reporting Summary linked to this article.

## Data availability

Raw embryo scRNAseq and hESC RNAseq produced for this study have been deposited in the GEO database under accession codes GSE250613, GSE250614, GSE297052, GSE297478, GSE239843. Processed scRNAseq data is available at Zendo repository 15128175. All PBAT-seq data generated in naïve hESCs are protected due to data privacy laws and are available upon request from the European Genome-phenome Archive EGAD50000001475. Raw confocal microscopy images generated in this study are available at Figshare 28597145. Previously published single-cell RNA-seq data from human embryos were downloaded from GEO accessions GSE36552 and GSE66507 and at EMBL-EBI ArrayExpress accession number E-MTAB-3929. Primed and naive hESC RNAseq datasets were downloaded from the ENA Browser https://www.ebi.ac.uk/ena/browser/home accessions PRJEB7132 [https://www.ebi.ac.uk/ena/browser/view/PRJEB7132], PRJNA522065, PRJNA575370, PRJEB12748, and PRJEB47485. All reagents, codes and materials used will be available from the corresponding author to any researcher for the purposes of reproducing or extending the analysis. Distribution of newly derived

naïve human embryonic stem cell lines requires permission from the UK Stem Cell Steering Committee and a completed Materials Transfer Agreement. Source data for this paper are available on Github[63,64]: 15640445 and 1512875 [https://doi.org/10.5281/zenodo.15128175].

## Code availability

Image analysis pipeline, code and source data for this paper are available on Github 15640445 and 1512875 [https://doi.org/10.5281/zenodo.15128175].

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

## Acknowledgements

We thank the donors of human embryos whose contributions were essential for this research. We thank all members of the Niakan lab for their technical assistance and for help and comments on the manuscripts. We thank the Loke Centre for Trophoblast Research for technical support and advice. We thank Florian Holfelder and Timo Kohler for support with confocal microscopy. We thank the Cambridge Stem Cell Institute for access to cell culture reagents. We thank the Francis Crick Institute's Science Technology Platforms: Lyn Healy and Liani Devito from the Human Embryo and Stem Cell Unit; Advanced Sequencing Facility, Advanced Light Microscopy and the Genomics Equipment Park. We thank the Babraham Institute's Genomics Facility and Bioinformatics Facility. Work in the laboratory of KKN was supported by the Wellcome 221856/Z/20/Z (KKN). Work in the laboratories of KKN and MH was supported by the Wellcome Human Developmental Biology Initiative 215116/Z/18/Z. Work in the laboratory of KKN was also supported by the Francis Crick Institute which receives its core funding from Cancer Research UK CC2074, the Medical Research Council CC2074 and Wellcome CC2074. Work in the JMB laboratory was supported by Lundbeck Foundation (R198-2015-412, R370-2021-617 and R400-2022-769), Independent Research Fund Denmark (DFF-8020-00100B, DFF-0134-00022B and DFF-2034-00025B) and the Danish National Research Foundation (DNRF116). The Novo Nordisk Foundation Center for Stem Cell Medicine (reNEW) is supported by the Novo Nordisk Foundation, grant number NNF21CC0073729 and previously NNF17CC0027852. Work in the CH laboratory as supported by the Wellcome Trust and Royal Society Sir Henry Dale Fellowship awarded to C.W.H. (222582/Z/21/Z). For the purpose of Open Access, the authors have applied a CC BY public copyright licence to any Author Accepted Manuscript version arising from this submission.

## Author contributions

Conceptualization: CSS; Methodology: CSS, AM, KKN; Investigation: CSS, AM, AF, KKN, DS, QH, GL, MLA; Visualisation: CSS, KKN; Analysis: CSS, KKN, LW, AF, MP, NS, GL, MLA, JMB; Funding acquisition: KKN, MH, JMB, CWH; Project administration: CSS, KKN; Supervision: CSS, KKN, CWH, MH, JMB; Human embryo consenting and facilitating donations: AP, KE, PS, LC, PG, VS, MT, MH, MC; Writing – original draft: CSS, KKN; Writing – review & editing: CSS, KKN with support from all authors.

## Funding

## Competing interests

The authors declare no competing interests.
