## [Transparent Peer Review file · Nature Communications]

Suppression of ERK signalling promotes pluripotent epiblast in the human blastocyst

Corresponding Author: Professor Kathy Niakan

Version 0:

Reviewer comments:

Reviewer #1

(Remarks to the Author)

Cell fate determination is a critical step during early embryonic development. In this work, Simon et.al. dissected the cell fate determination of epiblast and hypoblast cells in the early human blastocyst and found that FGF and ERK signaling influences epiblast versus hypoblast cell differentiation. They show exogenous FGF signaling activity will specify hypoblast cells, while ERK signaling inhibition will lead to the expansion of epiblast cells. This work provides certain mechanistic insights and potential manipulation into cell fate determination in human blastocysts, which will be of interest to the field of stem cell and human reproduction. Here are some concerns and suggestions that I think the authors may want to address to make the manuscript suitable for publication.

Majors:

1. This manuscript targets a controversial issue about cell fate decisions in the human blastocyst since previous studies showed the opposite conclusion regarding the function of the FGF/ERK pathway. Although the authors discussed some potential reasons for the differences, I would expect more pieces of evidence to support them: 1) whether the hypoblast specification of FGF induction could be blocked by the ERK inhibition, which will address the concern that previous works didn't block the entire pathway; 2) although the authors showed overall stable numbers of ICM cells under different FGF treatments (Fig. S1b), the total cell number of the blastocyst seemed decreased significantly (Fig. 1a). The authors should assess whether the observation was related to embryo viability or development age.

2. The authors used single-cell RNA-seq (SMART-seq) to assess the pluripotency of ERKi ICM cells. However, there were multiple problems regarding the quality and processing of the data.

1) the quality metrics (cell numbers, sequencing depth, gene numbers per cell, etc) of the data were not shown. The overall quality of the generated data should be comparable to these references. This is important to make the comparison between several sources of scRNA-seq data solid, which might have lots of batch effects in terms of data quality.

2) The number of single cells from ICM was very limited (Fig 3a, 3b, only several cells for DMSO and ERKi treatment). I would expect the TE cells to be excluded and the majority of the cells were ICM cells as the authors mentioned they manually collected the single cells. Limited cell numbers make the differential analysis in Fig. 3d and 3f very questionable (for instance, a majority of the genes were differential genes, which is not reasonable). In line with these concerns, Fig 3g didn't show any terms enriched with pluripotency.

With these concerns, I would suggest the authors carefully interpret these data or add more cells to make the conclusion solid.

3. Epigenetic marker like DNA methylation is significantly distinguished between naïve and primed hESCs. Therefore, it would be great to check the DNA methylation status of the derived hESCs in this work to add an additional layer of evidence.

4. The authors claimed that suppression of ERK signaling promotes pluripotent epiblast identity. However, the efficiency of hESC derivation without ERKi in Figs S4a-c (5 out of 8) was comparable or even slightly higher than this with 36h of ERKi (5 out of 9, Fig S4d and 4a). Moreover, replacing PXGL with UXGL medium also didn't improve the efficiency (Fig. s4g, 2/4 compared to 5/9 in Fig. 4a). With these considerations, it's hard to conclude that ERKi could promote pluripotency. More evidence or explanation should be provided to make this conclusion.

Minors:

1. What's the physiological concentration of FGF4 in human blastocyst? Whether the concentrations used in Fig 1 damage the growth of the blastocyst?
2. The markers to indicate epiblast and hypoblast cells in Fig 1 and Fig 2 were not consistent. Not sure the reason for that.
3. Fig 3b: the cell types of the clusters should be annotated, which is a standard way to help understand which cell type to choose in this paper.
4. Fig.3d: How many or what percentage of the genes were differentially expressed? What are the top differentially expressed genes and their functions?
5. The data filtering criteria were too loose to filter cells (i.e. > 200 genes/cell on Page 20). SMART-seq should give many more genes per cell.

Reviewer #2

(Remarks to the Author)

The manuscript by Simon et al re-examines the effect of FGF/ERK signalling on epiblast and hypoblast differentiation in human preimplantation embryos. Some experiments were carried out on other mammals for comparison purposes, and the impact on the derivation of naive hESCs was also verified.

Several years after the first publications reporting no effect, the authors clearly demonstrate that the balance between epiblast and hypoblast cell specification depends on the FGF/ERK pathway. These are therefore very important data. The data are clearly presented, the work fully supports the conclusions with adequate experiments.

However, although it is important to clarify the situation, the results are a thorough confirmation of previous findings. Indeed, the same laboratory showed that 1000 ng/ml FGF4 decreases NANOG expression (Fig 4f, Wamaitha et al, Nat Comm 2020), although there was no quantification at this dose. Furthermore, although these are in vitro systems, reports, some of which include the authors, demonstrate the importance of the FGF/ERK pathway for hypoblast differentiation (Linneberg-Agerholm, Development 2019; Okubo, Nature 2024).

Single-cell RNAseq had not yet been performed, however this is very descriptive, confirming that ERK inhibition maintains cells at an early epiblast stage. Unfortunately, the authors did not exploit the data to go further in understanding epiblast/hypoblast differentiation, to bring something really new.

Key points:

- 1000 ng/ml FGF decreases severely the number of ICM cells (cell death?), so cell identity cannot be assessed under these conditions. Thus, the lower number of epiblast cells in Fig1C should not be taken into account. It may be appropriate to delete the data at this dose of FGF from the main figure, or to modify the sentence in line 27.
- The use of 5-6 embryos per condition is too low, particularly in view of the high variability. Understandably, it's difficult to obtain human embryos, but this probably prevents significance from being achieved, and this is a very important point.
- fig 3h, supp fig 3f: it would be interesting to have the same graph with cell identity labels, to know where the 14 epiblast cells are.

Minor points:

- Supp Fig 1a legend: error for scRNAseq references
- Supp Fig1b: it is very difficult to distinguish the different shapes. It might be more appropriate to illustrate the different conditions with different colors (and shapes) for cell identity, although the groups can be easily identified.
- Is there an impact on the trophectoderm as has been reported in mice with regard to proliferation (Nichols, Development 2009)?

Version 1:

Reviewer comments:

Reviewer #1

(Remarks to the Author)

The authors have addressed my concerns. The manuscript is improved and suitable for publication now.

Reviewer #2

(Remarks to the Author)

In the revised version the manuscript has made significant advances, notably with the re-analyses of scRNAseq and novel findings with ERKi treated embryos showing differences between species (naive pluripotency in human versus dormant pluripotency in mouse).

A few points need to be completed/corrected:

- How the levels of pERK were quantified? The methods section refers to (Lea et al, Dev 2021) #16, however in this article only nuclear segmentation is described. How cell membrane was localised to segment the cytoplasm and quantify pERK?
- Fig 4h: how was defined a cell with an Epi-like identity, which markers(s)?
- could it be clearly identified which species are analysed in the panels (for example in supp Fig 4 there are panels with either mouse or human analyses)
- in supp Fig 1g-i and supp Fig 2c,d,h how many embryos were analysed?
- in supp Fig 1g-i and 2c-d what were the exact timing of FGF4 or ERKi administration in these other species (only indicated for bovine in methods), so that it can be repeated?
- for the mouse reference dataset, the indicated reference (#7 or #8) should be included in the main reference list
- supp fig 3: there are no j or k panels

- page 14 line 26: 1535 cells in the TE (compared with 83 Epiblast and 41 hypoblast)?

Version 2:

Reviewer comments:

Reviewer #2

(Remarks to the Author)

The authors have addressed my concerns

(Remarks on code availability)

REVIEWER COMMENTS

Reviewer #1 (Remarks to the Author):

Cell fate determination is a critical step during early embryonic development. In this work, Simon et.al. dissected the cell fate determination of epiblast and hypoblast cells in the early human blastocyst and found that FGF and ERK signaling influences epiblast versus hypoblast cell differentiation. They show exogenous FGF signaling activity will specify hypoblast cells, while ERK signaling inhibition will lead to the expansion of epiblast cells. This work provides certain mechanistic insights and potential manipulation into cell fate determination in human blastocysts, which will be of interest to the field of stem cell and human reproduction. Here are some concerns and suggestions that I think the authors may want to address to make the manuscript suitable for publication.

Majors:

1. This manuscript targets a controversial issue about cell fate decisions in the human blastocyst since previous studies showed the opposite conclusion regarding the function of the FGF/ERK pathway. Although the authors discussed some potential reasons for the differences, I would expect more pieces of evidence to support them: 1) whether the hypoblast specification of FGF induction could be blocked by the ERK inhibition, which will address the concern that previous works didn't block the entire pathway;

Previously, the use of inhibitors against FGFR and/or MEK did not inhibit endogenous signalling activity, as evidenced by no change in hypoblast or epiblast cell number (Roode et al. 2012, PMID: 22079695, Kuijk et. al., 2012, PMID: 22278923). In our study, we can block endogenous signalling activity using an ERK inhibitor, causing a significant reduction in hypoblast cells (Fig. 2g-e, Supplementary Data Fig. 2e-g).

The conclusions of our study, that the FGF/ERK pathway controls the epiblast vs hypoblast cell fate decision, have been corroborated by work published while our manuscript has been under review (Dattani et. al., 2024, PMID: 38823388). In this paper, they use a complementary approach to our study, using FGFR inhibitors on human blastoids and embryos. The concentration of FGFRi in the Dattani study is much higher than used previously, 0.5 μ M PD17 (FGFRi) compared to 0.5 μ M PD03 (MEKi) + 100nM PD17 (FGFRi), which had no effect on cell fate (Roode et. al.), providing evidence that the original studies didn't sufficiently block the FGF/MEK/ERK pathway.

In light of our work (on BioRxiv), and the Dattani study, the authors of the original Roode study have written a commentary piece offering technical explanations as to why their original experiments reached unjustified conclusions about the role of FGF/ERK pathway in hypoblast specification (Smith and Nichols 2024, PMID 38677582). We have referenced this commentary in the discussion. Together, this highlights the importance of our study in correcting the scientific record on a controversial issue about cell fate decision in the human blastocyst.

2) although the authors showed overall stable numbers of ICM cells under different FGF treatments (Fig. S1b), the total cell number of the blastocyst seemed decreased significantly (Fig. 1a). The authors should assess whether the observation was related to embryo viability or development age.

The reviewer is right to point out that the number of ICM cells in the 1000ng/ml FGF treatments are reduced when compared to controls, albeit not statistically significantly different. The overall embryo health as judged by morphology seems to be impacted at this high a concentration of FGF, independent of the starting embryo grade. At Reviewer 2's suggestion, we have removed experimental data that included the higher (1000ng/ml) FGF concentration and corresponding controls.

In addition, we have repeated the FGF treatments (for control, 250ng/ml, 500ng/ml and 750ng/ml FGF) to increase the sample number and improve the robustness of our findings (Fig. 1). Total sample size is now as follows: Control n = 11, 250ng/ml n = 8, 500ng/ml n = 9, 750ng/ml n=9. Following these updates, the % epiblast : hypoblast shows a statistically significant difference between all FGF treatment conditions and controls (Fig. 1d).

Reassuringly, the number of ICM cells (Supplementary Data Fig. 1c), number of TE cells (Supplementary Data Fig. 1d), and overall total number of cells (Supplementary Data Fig. S1e) at 250ng/ml, 500ng/ml and 750ng/ml FGF is not significantly different from that of control embryos, giving confidence that embryo viability is not impacted at these lower concentrations.

The results are included in an updated Fig. 1 and Supplementary Data Fig. 1 and agree with our initial results that exogenous FGF increase the % of hypoblast cells in the ICM. Full details are described in the main results text.

2. The authors used single-cell RNA-seq (SMART-seq) to assess the pluripotency of ERKi ICM cells. However, there were multiple problems regarding the quality and processing of the data.

1) the quality metrics (cell numbers, sequencing depth, gene numbers per cell, etc) of the data were not shown. The overall quality of the generated data should be comparable to these references. This is important to make the comparison between several sources of scRNA-seq data solid, which might have lots of batch effects in terms of data quality.

2) The number of single cells from ICM was very limited (Fig 3a, 3b, only several cells for DMSO and ERKi treatment). I would expect the TE cells to be excluded and the majority of the cells were ICM cells as the authors mentioned they manually collected the single cells. Limited cell numbers make the differential analysis in Fig. 3d and 3f very questionable (for instance, a majority of the genes were differential genes, which is not reasonable). In line with these concerns, Fig 3g didn't show any terms enriched with pluripotency.

With these concerns, I would suggest the authors carefully interpret these data or add more cells to make the conclusion solid.

To improve our data set we have repeated the single-cell RNAseq experiments in human embryos to add more cells. In addition, we have performed single-cell RNAseq in mouse embryos to compare to the results found in human.

We have performed more stringent thresholding for data quality, in line with published methods. This is detailed in the methods, and here for reference: 30% mitochondrial DNA, Identified genes > 8,000. After quality filtering we have n = 25 cells from the original dataset and n = 64 cells from the new experiments, a total of n = 89 cells. After clustering by cell identity, this corresponds to n = 53 TE cells, and n = 36 ICM cells.

Carry-over of trophoctoderm cells after immunosurgery is seen in other human single-cell RNAseq datasets (Petropoulos et al., 2016, PMID: 27062923) likely explained by internalisation

of polar trophectoderm (Corujo-Simon et al 2024, PMID: 38889726). We speculate that the long manipulation times during immunosurgery were having a negative impact on cell quality, therefore we performed the faster method of laser ablation (Blakeley et al. 2015, PMID: 26293300) to deplete trophectoderm cells in the most recent round of single-cell RNAseq. Both methods result in a depletion of trophectoderm cells, but not complete removal, and unfortunately once dissociated ICM vs trophectoderm cannot be readily distinguished by eye.

We have performed clustering analysis, and differential gene expression on our single-cell RNAseq dataset before integrating with published references. The new results are included in Figure 3 and Supplemental Figure 3, and discussed.

The key findings are that in ERKi treated embryos there is a (1) reduction in the number of hypoblast cells in agreement with IF studies (2) downregulation of key FGF targets within the epiblast associated with epiblast maturation, e.g. *ETVs* whilst maintenance and upregulation of naïve pluripotent genes (3) in non-epiblast lineages there is also a failure in maturation, and retention of pluripotency gene expression e.g. *NANOG*

3. Epigenetic marker like DNA methylation is significantly distinguished between naïve and primed hESCs. Therefore, it would be great to check the DNA methylation status of the derived hESCs in this work to add an additional layer of evidence.

We have included this as additional evidence, and the newly derived lines have hallmark hypomethylation characteristic of naïve hESC lines (Fig. 4i,j and Supplementary Fig. 5f,g)

4. The authors claimed that suppression of ERK signalling promotes pluripotent epiblast identity. However, the efficiency of hESC derivation without ERKi in Figs S4a-c (5 out of 8) was comparable or even slightly higher than this with 36h of ERKi (5 out of 9, Fig S4d and 4a). Moreover, replacing PXGL with UXGL medium also didn't improve the efficiency (Fig. s4g, 2/4 compared to 5/9 in Fig. 4a). With these considerations, it's hard to conclude that ERKi could promote pluripotency. More evidence or explanation should be provided to make this conclusion.

Suppression of ERK signalling promotes epiblast identity in the inner cell mass (ICM) at the expense of hypoblast identity. The proportion of ICM progenitor cells being specified to pluripotent epiblast is significantly higher upon ERKi treatment (Fig. 2g), generating an all-epiblast ICM in the majority of embryos (Supplementary Data Fig. 2g). These ERKi epiblast cells express both naïve and core pluripotent markers. At the protein level, we show *NANOG* (Fig 3d), *SOX2* (Supplementary Data Fig. 3h, and Fig. 4a), *OCT4* (Supplementary Data Fig. 3i), and *KLF17* (Fig. 4a) are expressed throughout the ICM in ERKi treated embryos. This is also validated at the RNA level in our scRNAseq, where ERKi epiblast cells express *NANOG*, *SOX2* (Fig. 3c), *KLF17* and *TDGF1* (Supplementary Data Fig. 3a). Differential gene expression also showed an upregulation of naïve markers *KLF2*, *KLF5* and *DNMT3L* in ERKi compared to control epiblast (Supplementary Data Fig. 3e). In addition, using a deep learning-model for lineage assignment, we find that ERKi epiblast more closely resemble more immature (or naïve) epiblast at Day 6 of human development, compared to control epiblast cells, that contain a greater proportion of more mature Day 7 and Late epiblast cells.

We aimed to functionally demonstrate pluripotent potential of these epiblast cells by directly deriving naïve hESCs. This would confirm if the ICM of ERKi embryos has been converted to a *bona fide* naïve pluripotent epiblast, rather than a confused or comprised cell state incapable of self-renewal in culture. We successfully derived and validated naïve hESC lines in both PXGL

and UXGL for ERKi treated embryos, showing that the ERKi epiblast is truly pluripotent. We are very careful to not make any claims in manuscript about the efficiency of derivation between ERKi and control embryos, or PXGL and UXGL, because of the low number of embryos and lines derived.

Taken together, our data demonstrate that ERKi promotes specification within the ICM to an epiblast identity, which both express pluripotent markers and can generate naïve hESC lines. We have edited the text to improve the clarity of this explanation in the main results and discussion.

Minors:

1. What's the physiological concentration of FGF4 in human blastocyst? Whether the concentrations used in Fig 1 damage the growth of the blastocyst?

The concentration of endogenous FGF4 production is not known and would be difficult to measure in human blastocysts.

In response to points both reviewers raised, we have removed the highest (1000ng/ml FGF) concentration as this seemed to negatively impact embryo viability, whereas there is good embryo viability at 250-750ng/ml FGF concentrations.

2. The markers to indicate epiblast and hypoblast cells in Fig 1 and Fig 2 were not consistent. Not sure the reason for that.

The markers used to define epiblast and hypoblast for FGF (Fig. 1a) and ERKi (Fig. 2d) treatments are the same. In both Fig. 1a and Fig. 2d, NANOG (epiblast) and GATA4 (hypoblast) are used to label late-stage blastocysts at Day 6.5. Therefore, the markers used to determine cell fate changes are consistent between treatment types.

Experiments using SOX2 (epiblast) and OTX2 (hypoblast) in Fig. 2a were performed alongside pERK staining, in early to late blastocyst (Day 5 to Day 6.5). OTX2 is an earlier hypoblast marker than Gata4 (Curujo-Simon et al 2023, PMID: 37102672) and was therefore used as a hypoblast lineage marker in experiments with earlier stage embryos, also necessitating a change in the epiblast marker due to species incompatibility between OTX2 and NANOG antibodies.

We have included this point in the main text to clarify the rationale of using different markers to readers.

3. Fig 3b: the cell types of the clusters should be annotated, which is a standard way to help understand which cell type to choose in this paper.

We appreciate the reviewer's point and have included this in a revised Fig. 3a and Fig. 3b.

4. Fig.3d: How many or what percentage of the genes were differentially expressed? What are the top differentially expressed genes and their functions?

The number of differentially expressed genes is now included in the volcano plots Fig. 3d and Supplementary Data Fig. 3d,f . Differentially expressed genes for each cell type are included in heatmaps, and the full list included in Supplementary Tables 1 and 2.

We have also performed gene set enrichment analysis on the DESeq2 results, and the most

highly enriched terms are shown in Supplementary Data Fig. 3c. The top enriched gene sets; oestrogen response, mTOR signalling, and cholesterol and fatty acid metabolism, were all upregulated in ERKi epiblast cells. These biological processes are related to lipid metabolism, a hallmark of naïve pluripotency in human ESCs and embryos, and suggest a shift in bioenergetic requirements of ERKi epiblast to a more naïve-like pluripotent state.

5. The data filtering criteria were too loose to filter cells (i.e. > 200 genes/cell on Page 20). SMART-seq should give many more genes per cell.

We have changed the data filtering thresholds (to only include cells with > 8,000 genes per cell and <30% mitochondrial reads) and optimised our experimental methods in the newest batch of single-cell RNAseq to improve data quality.

Reviewer #2 (Remarks to the Author):

The manuscript by Simon et al re-examines the effect of FGF/ERK signalling on epiblast and hypoblast differentiation in human preimplantation embryos. Some experiments were carried out on other mammals for comparison purposes, and the impact on the derivation of naive hESCs was also verified.

Several years after the first publications reporting no effect, the authors clearly demonstrate that the balance between epiblast and hypoblast cell specification depends on the FGF/ERK pathway. These are therefore very important data.

The data are clearly presented, the work fully supports the conclusions with adequate experiments.

However, although it is important to clarify the situation, the results are a thorough confirmation of previous findings. Indeed, the same laboratory showed that 1000 ng/ml FGF4 decreases NANOG expression (Fig 4f, Wamaitha et al, Nat Comm 2020), although there was no quantification at this dose. Furthermore, although these are in vitro systems, reports, some of which include the authors, demonstrate the importance of the FGF/ERK pathway for hypoblast differentiation (Linneberg-Agerholm, Development 2019; Okubo, Nature 2024).

In Wamaitha et al. (Nature Communications, 2020), human embryo treatments with FGF2 at 1000 ng/ml (and 100 ng/ml) from Day 2 to Day 6.5 resulted in a loss of NANOG+ cells. Although there were GATA6+ cells present, GATA6 marks both the trophectoderm and hypoblast. Additionally, cell numbers were never quantified, and at both high and low concentrations of FGF2, the embryos were much smaller compared to controls. Therefore, from these experiments, we could not determine whether the observed effects were due to a cell-fate conversion of epiblast to hypoblast, an impact on embryo viability, or a failure in the initial ICM vs TE specification. In the Wamaitha paper fig. 4g,h, focusing on a shorter time window Day 5 – Day 6.5 (as in this current study), 100 ng/ml FGF2 treatment did not cause a significant change in the number of SOX17+ hypoblast or NANOG+ epiblast cells.

Therefore, to address the question of whether FGF/ERK controls epiblast vs hypoblast fate our current study has important distinctions from previous work:

- (1) The use of FGF4, and not FGF2, to reflect ligand expression in the human embryo (Supp. Fig. 1a).
- (2) Dosage response from low to high concentrations of exogenous FGF4 ligand (250ng/ml – 750ng/ml)
- (3) Treatments from Day 5 – Day 6.5 encompassing the window of epiblast vs hypoblast specification

- (4) Use of GATA4 as a hypoblast specific marker
- (5) Inhibition of the FGF pathway using ERKi, showing the FGF/ERK is both necessary and sufficient for cell fate conversion within the ICM.
- (6) Single-cell RNAseq analysis
- (7) Derivation of naïve hESC to demonstrate the functional capacity of ERKi treated embryos to retain naïve pluripotency

Single-cell RNAseq had not yet been performed, however this is very descriptive, confirming that ERK inhibition maintains cells at an early epiblast stage. Unfortunately, the authors did not exploit the data to go further in understanding epiblast/hypoblast differentiation, to bring something really new.

In the revised version of this manuscript we have made extensive changes to single-cell RNAseq analysis, collecting additional human embryo samples, and including a comparative study with ERKi mouse embryos.

In our updated single-cell analysis the key findings are that in human ERKi treated embryos:
(1) reduction in the number of hypoblast cells in agreement with IF studies
(2) downregulation of key FGF targets within the epiblast associated with epiblast maturation, e.g. *ETVs* whilst maintenance and upregulation of naïve pluripotent genes
(3) in non-epiblast lineages there is also a failure in maturation, and retention of pluripotency gene expression e.g. *NANOG*.

In addition, we find that while ERK inhibition biases ICM fate toward epiblast in both mouse and human, its downstream effect on lineage progression differs. In the human, ERKi epiblast retains a naïve pluripotent signature, whereas the mouse ERKi epiblast exhibits a dormant pluripotent state, indicating divergent functions of ERK in the epiblast and pluripotency between the species.

Furthermore, we derive naïve human embryonic stem cells directly from ERKi embryos, functionally demonstrating that in human, ERK inhibition maintains epiblast cells in a naïve pluripotent state.

Key points:

- 1000 ng/ml FGF decreases severely the number of ICM cells (cell death?), so cell identity cannot be assessed under these conditions. Thus, the lower number of epiblast cells in Fig1C should not be taken into account. It may be appropriate to delete the data at this dose of FGF from the main figure, or to modify the sentence in line 27.
- The use of 5-6 embryos per condition is too low, particularly in view of the high variability. Understandably, it's difficult to obtain human embryos, but this probably prevents significance from being achieved, and this is a very important point.

The reviewer is right to point out that the number of ICM cells in the 1000ng/ml FGF treatments are reduced when compared to controls, albeit not statistically significantly different. The overall embryo health as judged by morphology seems to be impacted at this high a concentration of FGF, independent of the starting embryo grade. At Reviewer 2's suggestion, we have removed experimental data that included the higher (1000ng/ml) FGF concentration and corresponding controls.

In addition, we have repeated the FGF treatments (for control, 250ng/ml, 500ng/ml and 750ng/ml FGF) to increase the sample number and improve the robustness of our findings (Fig.

1). Total sample size is now as follows: Control n = 11, 250ng/ml n = 8, 500ng/ml n = 9, 750ng/ml n=9. Following these updates, the % epiblast : hypoblast shows a statistical significant difference between all FGF treatment conditions and controls (Fig. 1d).

Reassuringly, the number of ICM cells (Supplementary Data Fig. 1c), number of TE cells (Supplementary Data Fig. 1d), and overall total number of cells (Supplementary Data Fig. S1e) at 250ng/ml, 500ng/ml and 750ng/ml FGF is not significantly different from that of control embryos, giving confidence that embryo viability is not impacted at these lower concentrations.

The results are included in an updated Fig. 1 and Supplementary Data Fig. 1 and agree with our initial results that exogenous FGF increase the % of hypoblast cells in the ICM. Full details are described in the main results text.

- fig 3h, supp fig 3f: it would be interesting to have the same graph with cell identity labels, to know where the 14 epiblast cells are.

In response to Reviewer 1 we have performed additional single-cell RNAseq experiments to increase the n number. We have revised these figures in new Fig. 3 and Supplementary Fig. 3. Cell identity labels are included in Fig. 3b.

Minor points:

- Supp Fig 1a legend: error for scRNAseq references

Thank you for pointing out this mistake, we have updated the reference.

- Supp Fig1b: it is very difficult to distinguish the different shapes. It might be more appropriate to illustrate the different conditions with different colors (and shapes) for cell identity, although the groups can be easily identified.

We have enlarged the graph, size of points and increased the opacity to help improve readability of Supplementary Fig. 1b

- Is there an impact on the trophectoderm as has been reported in mice with regard to proliferation (Nichols, Development 2009)?

We thank the reviewer for raising this, and have included analysis of both FGF and ERKi treatments on the number of trophectoderm cells in the revision (Supplementary Fig. 1d and Supplementary Data Fig. 2j). In ERKi treatments there is an impact on trophectoderm proliferation, in line with observations in mouse (Nichols et al 2009 Development). We have updated the text to include these results:

ERKi treated embryos form hatching, expanded blastocysts by Day 6.5, with GATA3+ trophectoderm (Fig. 2d), indicating that formation of the trophectoderm, unlike the hypoblast, is not dependent on ERK signalling. However, ERKi embryos do have significantly fewer number of trophectoderm cells than controls (Supplementary Data Fig. 2j, $p = 0.04$) reducing the overall total embryo cell numbers (Supplementary Data Fig. 2k), suggesting a conserved role of ERK signalling in the proliferation of the human trophectoderm, similar to mouse¹².

Reviewer response: point by point

*Simon et. al.,
Suppression of ERK signalling promotes pluripotent epiblast in the human blastocyst.
NCOMMS-24-04269A*

REVIEWER COMMENTS

Reviewer #1 (Remarks to the Author):

The authors have addressed my concerns. The manuscript is improved and suitable for publication now.

Reviewer #2 (Remarks to the Author):

- How the levels of pERK were quantified? The methods section refers to (Lea et al, Dev 2021) #16, however in this article only nuclear segmentation is described. How cell membrane was localised to segment the cytoplasm and quantify pERK?

To quantify the nuclear versus cytoplasmic levels of pERK, nuclear segmentation was expanded by 4 pixels to generate a cytoplasmic ring around the nucleus and the ratio of nuclear or cytoplasmic expression was determined. The Gitlab repository includes further step-by-step details of the CellProfiler pipelines used. We have updated the materials and methods section to include this statement.

- Fig 4h: how was defined a cell with an Epi-like identity, which markers(s)?

From the Materials and Methods Human ESC RNA-seq analysis section we have clarified that: "Similarity to an epiblast identity was assessed using DeconRNASeq (58), based on a previous publication (34), using as a reference set high confidence epiblast cells that were obtained from previous work (59)."

Details of the "epiblast" classification are in Ref 59: Alanis-Lobato et al 2023 Life Sci Alliance (PMID: 37879938).

- could it be clearly identified which species are analysed in the panels (for example in supp Fig 4 there are panels with either mouse or human analyses)

We thank the reviewer for this suggestion. Supplementary Fig 4 is exclusively mouse data. We have updated the Supplementary Fig. 4. title to "Supplementary Fig. 4. Mouse single-cell transcriptomics reveals conserved ERK function in hypoblast specification but divergence in pluripotent epiblast lineage maintenance". We have also updated the figure legend to make it clear that the panels refer to mouse data.

- in supp Fig 1g-i and supp Fig 2c,d,h how many embryos were analysed?

We thank the reviewer for this comment which has prompted us to amend the figure legends with the sample sizes. Below is a list of the number of embryos analysed in each figure.

Supp. Fig. 1g (mouse FGF): Control n = 6, FGF n = 10

Supp. Fig. 1h (rat FGF): Control n = 10, FGF n = 10
Supp. Fig. 1i (cow FGF): Control n = 3, FGF n=3
Supp fig. 2c (mouse ERKi): Control n = 5, ERKi n=4
Supp fig 2d (cow ERKi): Control n = 3, ERKi n=3
Supp Fig. 2h (human PDGFRA): Control n = 4, ERKi n=4

- in supp Fig 1g-i and 2c-d what were the exact timing of FGF4 or ERKi administration in these other species (only indicated for bovine in methods), so that it can be repeated?

We thank the reviewer for this suggestion and have further clarified in the figure legend and the supplementary methods section the following information:

Supp Fig 1g: Confocal images of mouse embryos following either 1000ng/ml FGF4 plus Heparin or control treatment from day 2.5 morula stage for 48 hrs until day 4.5 late blastocyst stage.

Supp Fig 1h: Confocal images of rat embryos following either 1000ng/ml FGF4 plus Heparin or control treatment from day 3.5 early blastocyst stage for 24 hrs until day 4.5 late blastocyst stage.

Supp Fig. 1i: Confocal images of cow embryos following either 750ng/ml FGF4 plus Heparin or control treatment from day 7 morula stage for 48 hrs until day 9 blastocyst stage.

Supp Fig 2c: Confocal images of mouse embryos following either ERKi or control treatment from day 2.5 morula stage for 48 hrs until day 4.5 late blastocyst stage.

Supp Fig 2d: Confocal images of cow embryos following either ERKi or control treatment from day 7 morula stage for 48 hrs until day 9 the blastocyst stage.

- for the mouse reference dataset, the indicated reference (#7 or #8) should be included in the main reference list

In the main reference list, references #27 and #28 are identical to references #7 and #8 in supplementary information and are therefore already included. We only used the Proks et al. 2025 mouse dataset in supplementary figure 4, which is noted in the figure legend and reference list of the supplementary information (reference #8).

- supp fig 3: there are no j or k panels

We thank the reviewer for pointing out this error. We have removed the text related to these previous panels from the Supplementary Fig 3 legend.

- page 14 line 26: 1535 cells in the TE (compared with 83 Epiblast and 41 hypoblast)?

TE cells greatly outnumber ICM cells in human embryos and the numbers reflect the proportion of cells in human embryos expressing markers associated with each lineage.